# ALIGN-RUDDER: LEARNING FROM FEW DEMONSTRATIONS BY REWARD REDISTRIBUTION

## ABSTRACT

Reinforcement Learning algorithms require a large number of samples to solve complex tasks with sparse and delayed rewards. Complex tasks are often hierarchically composed of sub-tasks. A step in the $Q$-function indicates solving a sub-task, where the expectation of the return increases. RUDDER identifies these steps and then redistributes reward to them, thus immediately giving reward if sub-tasks are solved. Since the delay of rewards is reduced, learning is considerably sped up. However, for complex tasks, current exploration strategies struggle with discovering episodes with high rewards. Therefore, we assume that episodes with high rewards are given as demonstrations and do not have to be discovered by exploration. Typically the number of demonstrations is small and RUDDER's LSTM model does not learn well. Hence, we introduce Align-RUDDER, which is RUDDER with two major modifications. First, Align-RUDDER assumes that episodes with high rewards are given as demonstrations, replacing RUDDER's safe exploration and lessons replay buffer. Second, we substitute RUDDER's LSTM model by a profile model that is obtained from multiple sequence alignment of demonstrations. Profile models can be constructed from as few as two demonstrations. Align-RUDDER inherits the concept of reward redistribution, which speeds up learning by reducing the delay of rewards. Align-RUDDER outperforms competitors on complex artificial tasks with delayed reward and few demonstrations. On the MineCraft `ObtainDiamond` task, Align-RUDDER is able to mine a diamond, though not frequently.

## 1 INTRODUCTION

**Overview of our method, Align-RUDDER.** Reinforcement learning algorithms struggle with learning complex tasks that have sparse and delayed rewards (Sutton & Barto, 2018; Rahmandad et al., 2009; Luoma et al., 2017). RUDDER (Arjona-Medina et al., 2019) has shown to excel for learning sparse and delayed rewards. RUDDER requires episodes with high rewards, to store them in its lessons replay buffer for learning. However, for complex tasks episodes with high rewards are difficult to find by current exploration strategies. Humans and animals obtain high reward episodes by teachers, role models, or prototypes. In this context, we assume that episodes with high rewards are given as demonstrations. Consequently, RUDDER's safe exploration and lessons replay buffer can be replaced by these demonstrations. Generating demonstrations is often tedious for humans and time-consuming for automated exploration strategies, therefore typically only few demonstrations are available. However, RUDDER's LSTM as a deep learning method requires many examples for learning. Therefore, we replace RUDDER's LSTM by a profile model obtained from multiple sequence alignment of the demonstrations. Profile models are well known in bioinformatics, where they are used to score new sequences according to their sequence similarity to the aligned sequences. RUDDER's LSTM predicts the return for an episode given a state-action sub-sequence. Our method replaces the LSTM by the score of this sub-sequence if aligned to the profile model. In the RUDDER implementation, the LSTM predictions are used for return decomposition and reward redistribution by using the difference of consecutive predictions. Align-Rudder performs return decomposition and reward redistribution via the difference of alignment scores for consecutive sub-sequences.

**Align-RUDDER vs. temporal difference and Monte Carlo.** We assume to have high reward episodes as demonstrations. Align-RUDDER uses these episodes to identify through alignment

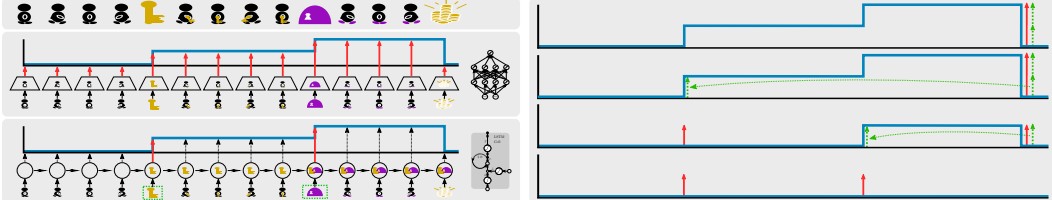

Figure 1: **Basic insight of RUDDER (left panel) and reward redistribution (right panel). Left panel, row 1**: An agent has to take a key to open the door to a treasure. Both events increase the probability of obtaining the treasure. **row 2**: Learning the $Q$-function by a fully connected network requires to predict the expected return from every state-action pair (red arrows). **row 3**: Learning the $Q$-functions by memorizing relevant state-action pairs requires only to predict the steps (red arrows). **Right panel, row 1**: The $Q$-function is the future expected reward (blue curve) since reward is given only at the end. The $Q$-function is a step function, where green arrows indicate steps and the big red arrow the expected return. **row 2**: The redistributed reward (small red arrow) removes a step in the $Q$-function and the future expected return becomes constant at this step (blue curve). **row 3**: After redistributing the return, the expected future reward is equal to zero (blue curve at zero). Learning focuses on the expected immediate reward (red arrows) since delayed rewards are no longer present.

techniques state-actions that are indicative for high rewards. Align-RUDDER redistributes rewards to these state-actions to adjust a policy so that these state-actions are reached more often. Consequently, the return is increased and relevant episodes are sampled more frequently. For *delayed rewards* and *model-free* reinforcement learning: (I) temporal difference (TD) suffers from vanishing information (Arjona-Medina et al., 2019); (II) Monte Carlo (MC) averages over all futures, leading to high variance (Arjona-Medina et al., 2019). Monte-Carlo Tree Search (MCTS) which was used for Go and chess is a model-based method that can handle delayed and rare rewards (Silver et al., 2016; 2017).

**Basic insight: $Q$-functions (action-value functions) are step functions.** Complex tasks are often hierarchically composed of sub-tasks (see Fig. 1). Hence, the $Q$-function of an optimal policy resembles a step function. A step is a change in return expectation, that is, the expected amount of the return or the probability to obtain the return changes. Steps indicate achievements, failures, accomplishing sub-tasks, or changes of the environment. To identify large steps in the $Q$-function speeds up learning, since it allows (i) to increase the return and (ii) to sample more relevant episodes.

A $Q$-function predicts the expected return from every state-action pair (see Fig. 1), which is prone to make a prediction error that hampers learning. Since the $Q$-function is mostly constant, it is not necessary to predict the expected return for every state-action pair. It is sufficient to *identify relevant state-actions* across the whole episode and use them for predicting the expected return (see Fig. 1). The LSTM network (Hochreiter & Schmidhuber, 1995; 1997a;b) can store the relevant state-actions in its memory cells. Subsequently, it only updates them if a new relevant state-action pair appears, when also the output changes which is otherwise constant. The basic insight that $Q$-functions are step functions is the motivation for identifying these steps via return decomposition and speeding up learning via reward redistribution.

**Reward Redistribution: Idea and return decomposition.** We consider reward redistributions that are obtained by return decomposition given an episodic Markov decision process (MDP). The $Q$-function is assumed to be a step function (blue curve, row 1 of Fig. 1, right panel). Return decomposition identifies the steps of the $Q$-function (green arrows in Fig. 1, right panel). A function (LSTM in RUDDER, alignment model in Align-RUDDER) predicts the expected return (red arrow, row 1 of Fig. 1, right panel) given the state-action sub-sequence. The prediction is decomposed into single steps of the $Q$-function (green arrows in Fig. 1). The redistributed rewards (small red arrows in second and third row of right panel of Fig. 1) remove the steps. Consequently, the expected future reward is equal to zero (blue curve at zero in last row in right panel of Fig. 1). Having future rewards of zero means that learning the $Q$-values simplifies to estimating the expected immediate rewards (small red arrows in right panel of Fig. 1), since delayed rewards are no longer present.

**Reward redistribution using multiple sequence alignment.** RUDDER uses an LSTM model for reward redistribution via return decomposition. The reward redistribution is the difference of two subsequent predictions of the LSTM model. If a state-action pair increases the prediction of the return,

Figure 2: Left: Alignment of biological sequences (triosephosphate isomerase) giving a conservation score. Right: Alignment of demonstrations using the conservation score for reward redistribution.

then it is immediately rewarded. Using state-action sub-sequences $(s,a)_{0:t} = (s_0, a_0, \ldots, s_t, a_t)$, the redistributed reward is $R_{t+1} = g((s,a)_{0:t}) - g((s,a)_{0:t-1})$, where $g$ is the return decomposition function, which is an LSTM model that predicts the return of the episode. The LSTM model first learns to approximate the largest steps of the $Q$-function, since they reduce the prediction error the most. Therefore the LSTM model extracts first the relevant state-actions pairs (events).

We now use techniques from sequence alignment to replace the LSTM model by a profile model for return decomposition. The profile model is the result of a multiple sequence alignment of the demonstrations and allows aligning new sequences to it. Both the sub-sequences $(s,a)_{0:t-1}$ and $(s,a)_{0:t}$ are mapped to sequences of events and then are aligned to the profile model. Thus, both sequences receive an alignment score $S$, which is proportional to the function $g$. The redistributed reward is again $R_{t+1} = g((s,a)_{0:t}) - g((s,a)_{0:t-1})$ (see Eq. (3)). Fig. 2 shows an alignment of biological sequences and an alignment of demonstrations. The concepts of reward redistribution and return decomposition are reviewed in Sec. 2 and reward redistribution by sequence alignment is explained in Sec. 3.

**Related work.** Learning from demonstrations has been widely studied over the last 50 years (Billard et al., 2008). The most prominent example is imitation learning, which profits from supervised techniques when the number of available demonstrations is large enough (Michie et al., 1990; Pomerleau, 1991; Michie & Camacho, 1994; Schaal, 1996; Kakade & Langford, 2002). However, policies trained with imitation learning tend to drift away from demonstration trajectories due to a distribution shift (Ross & Bagnell, 2010). This effect can be mitigated (III et al., 2009; Ross & Bagnell, 2010; Ross et al., 2011; Judah et al., 2014; Sun et al., 2017; 2018). Many approaches use demonstrations for initialization, e.g. of policy networks (Taylor et al., 2011; Silver et al., 2016; Rajeswaran et al., 2018; Le et al., 2018), value function networks (Hester et al., 2017; 2018), both networks (Zhang & Ma, 2018; Nair et al., 2018), or an experience replay buffer (Hosu & Rebedea, 2016; Vecerík et al., 2017). Beyond initialization, demonstrations are used to define constraints (Kim et al., 2013), generate sub-goals (Eysenbach et al., 2019), enforce regularization (Reddy et al., 2020), guide exploration (Subramanian et al., 2016; Jing et al., 2019), or shape rewards (Judah et al., 2014; Brys et al., 2015; Suay et al., 2016). Demonstrations may serve for inverse reinforcement learning (Ng & Russell, 2000; Abbeel & Ng, 2004; Ho & Ermon, 2016), which aims at learning a (non-sparse) reward function that best explains the demonstrations. Learning reward functions requires a large number of demonstrations (Syed & Schapire, 2007; Ziebart et al., 2008; Tucker et al., 2018; Silva et al., 2019). Some approaches rely on few-shot or/and meta learning (Duan et al., 2017; Finn et al., 2017; Zhou et al., 2020). Meta learning has been used for images to enable inference in a latent state model to acquire new skills given observations and rewards (Zhao et al., 2020). Like with supervised meta learning, a reinforcement model was pre-trained on a large batch of fixed, pre-collected data (Mitchell et al., 2020). However, few-shot and meta learning demand a large set of auxiliary tasks or prerecorded data. Concluding, most methods that learn from demonstrations rely on the availability of many demonstrations (Khardon, 1999; Lopes et al., 2009), in particular, if using deep learning methods (Bengio & Lecun, 2007; Lakshminarayanan et al., 2016). Some methods can learn on few demonstrations like Soft $Q$ Imitation Learning (SQIL) (Reddy et al., 2020), Generative Adversarial Imitation Learning (GAIL) (Ho & Ermon, 2016), and Deep $Q$-learning from Demonstrations (DQfD) (Hester et al., 2018).

Align-RUDDER allows to identify sub-goals and sub-tasks and assigns reward to them, therefore it is related to hierarchical reinforcement learning. Hierarchical reinforcement learning has been formally investigated by the seminal papers on the option framework (Sutton et al., 1999) and on the MAXQ framework (Dieterich, 2000). Also the recursive composition of option models into other option

models formally treats hierarchical reinforcement learning (Silver & Ciosek, 2012). However, these methods do not address the problem of finding good options, good sub-goals, or good sub-tasks. Good options are found by constructing random tasks, solving them, and identifying frequently observed states as targets (Stolle & Precup, 2002). Gradient-based approaches have been used for improving the termination function for options (Comanici & Precup, 2010) and for a particular structure of the initiation sets and termination functions (Mankowitz et al., 2016). A unified policy consisting of intra-option policies, option termination conditions, and an option selection policy (inter options) is optimized by standard policy gradient algorithms (Levy & Shimkin, 2012). Parametrized options are learned by treating the termination functions as hidden variables and using expectation maximization for leaning (Daniel et al., 2016). The DQN framework is used to implement a gradient-based option learner, which uses intrinsic rewards to learn the internal policies of options, and extrinsic rewards to learn the policy over options (Kulkarni et al., 2016). Options have been jointly learned with an associated policy using the policy gradient theorem for options (Bacon et al., 2017). A slow time-scale manager module learns sub-goals that are achieved by fast time-scale worker module (Vezhnevets et al., 2017).

## 2   REVIEW REWARD REDISTRIBUTION

Reward redistribution and return decomposition are concepts introduced in RUDDER and are also fundamental to Align-RUDDER. Reward redistribution based on return decomposition eliminates – or at least mitigates – delays of rewards while preserving the same optimal policies. Align-RUDDER is justified by the theory of return decomposition and reward redistribution when using multiple sequence alignment for constructing a reward redistribution model. In this section, we briefly review the concepts and theory of return decomposition and reward redistribution.

**Preliminaries.** We consider a finite MDP defined by the 5-tuple $\mathcal{P} = (\mathcal{S}, \mathcal{A}, \mathcal{R}, p, \gamma)$ where the state space $\mathcal{S}$ and the action space $\mathcal{A}$ are sets of finite states $s$ and actions $a$ and $\mathcal{R}$ the set of bounded rewards $r$. For a given time step $t$, the corresponding random variables are $S_t, A_t$ and $R_{t+1}$. Furthermore, $\mathcal{P}$ has transition-reward distributions $p(S_{t+1} = s', R_{t+1} = r \mid S_t = s, A_t = a)$, and a discount factor $\gamma \in (0, 1]$, which we keep at $\gamma = 1$. A Markov policy $\pi(a \mid s)$ is a probability of an action $a$ given a state $s$. We consider MDPs with finite time horizon or with an absorbing state. The discounted return of a sequence of length $T$ at time $t$ is $G_t = \sum_{k=0}^{T-t} \gamma^k R_{t+k+1}$. As usual, the $Q$-function for a given policy $\pi$ is $q^\pi(s, a) = \mathrm{E}_\pi[G_t \mid S_t = s, A_t = a]$. $\mathrm{E}_\pi[x \mid s, a]$ is the expectation of $x$, where the random variable is a sequence of states, actions, and rewards that is generated with transition-reward distribution $p$, policy $\pi$, and starting at $(s, a)$. The goal is to find an optimal policy $\pi^* = \mathrm{argmax}_\pi \mathrm{E}_\pi[G_0]$ maximizing the expected return at $t = 0$. We assume that the states $s$ are time-aware (time $t$ can be extracted from each state) in order to assure stationary optimal policies. According to Proposition 4.4.3 in (Puterman, 2005), a deterministic optimal policy $\pi^*$ exists.

**Definitions.** A *sequence-Markov decision process* (SDP) is defined as a decision process that has Markov transition probabilities but a reward probability that is not required to be Markov. Two SDPs $\tilde{\mathcal{P}}$ and $\mathcal{P}$ with different reward probabilities are *return-equivalent* if they have the same expected return at $t = 0$ for each policy $\pi$, and *strictly return-equivalent* if they additionally have the same expected return for every episode. Since for every $\pi$ the expected return at $t = 0$ is the same, return-equivalent SDPs have the same optimal policies. A *reward redistribution* is a procedure that —for a given sequence of a delayed reward SDP $\tilde{\mathcal{P}}$— redistributes the realization or expectation of its return $\tilde{G}_0$ along the sequence. This yields a new SDP $\mathcal{P}$ with $R$ as random variable for the redistributed reward and the same optimal policies as $\tilde{\mathcal{P}}$:

**Theorem 1** (Arjona-Medina et al. (2019)). *Both the SDP $\tilde{\mathcal{P}}$ with delayed reward $\tilde{R}_{t+1}$ and the SDP $\mathcal{P}$ with redistributed reward $R_{t+1}$ have the same optimal policies.*

The delay of rewards is captured by the *expected future rewards* $\kappa(m, t-1)$ at time $(t-1)$. $\kappa$ is defined as $\kappa(m, t-1) := \mathrm{E}_\pi[\sum_{\tau=0}^m R_{t+1+\tau} \mid s_{t-1}, a_{t-1}]$, that is, at time $(t-1)$ the expected sum of future rewards from $R_{t+1}$ to $R_{t+1+m}$ but not the immediate reward $R_t$. A reward redistribution is defined to be *optimal*, if $\kappa(T - t - 1, t) = 0$ for $0 \leqslant t \leqslant T - 1$, which is equivalent to $\mathrm{E}[R_{t+1} \mid s_{t-1}, a_{t-1}, s_t, a_t] = \tilde{q}^\pi(s_t, a_t) - \tilde{q}^\pi(s_{t-1}, a_{t-1})$:

**Theorem 2** (Arjona-Medina et al. (2019)). *We assume a delayed reward MDP $\tilde{\mathcal{P}}$, with episodic reward. A new SDP $\mathcal{P}$ is obtained by a second order Markov reward redistribution, which ensures*

*that $\mathcal{P}$ is return-equivalent to $\tilde{\mathcal{P}}$. For a specific $\pi$, the following two statements are equivalent:*
*(I)  $\kappa(T-t-1,t) = 0$, i.e. the reward redistribution is optimal,*

$$(II) \ \ \mathrm{E}\left[R_{t+1} \mid s_{t-1}, a_{t-1}, s_t, a_t\right] \ = \ \tilde{q}^{\pi}(s_t, a_t) \ - \ \tilde{q}^{\pi}(s_{t-1}, a_{t-1}) \,. \tag{1}$$

*An optimal reward redistribution fulfills for $1 \leqslant t \leqslant T$ and $0 \leqslant m \leqslant T - t$: $\kappa(m, t-1) = 0$.*

This theorem shows that an optimal reward redistribution relies on steps $\tilde{q}^{\pi}(s_t, a_t) - \tilde{q}^{\pi}(s_{t-1}, a_{t-1})$ of the $Q$-function. Identifying the largest steps in the $Q$-function detects the largest rewards that have to be redistributed, which makes the largest progress towards obtaining an optimal reward redistribution. If the reward redistribution is optimal, the $Q$-values of $\mathcal{P}$ are given by $q^{\pi}(s_t, a_t) = \tilde{q}^{\pi}(s_t, a_t) - \psi^{\pi}(s_t)$ and therefore $\tilde{\mathcal{P}}$ and $\mathcal{P}$ have the same advantage function:

**Theorem 3** (Arjona-Medina et al. (2019))**.** *If the reward redistribution is optimal, then the $Q$-values of the SDP $\mathcal{P}$ are $q^{\pi}(s_t, a_t) = r(s_t, a_t)$ and*

$$q^{\pi}(s_t, a_t) \ = \ \tilde{q}^{\pi}(s_t, a_t) \ - \ \mathrm{E}_{s_{t-1}, a_{t-1}}\left[\tilde{q}^{\pi}(s_{t-1}, a_{t-1}) \mid s_t\right] \ = \ \tilde{q}^{\pi}(s_t, a_t) \ - \ \psi^{\pi}(s_t) \,. \tag{2}$$

*The SDP $\mathcal{P}$ and the original MDP $\tilde{\mathcal{P}}$ have the same advantage function.*

For an optimal reward redistribution only the expectation of the immediate reward $r(s_t, a_t) = \mathrm{E}\left[R_{t+1} \mid s_t, a_t\right]$ must be estimated. This considerably simplifies learning.

**Learning methods according to Arjona-Medina et al. (2019).** The redistributed reward serves as reward for a subsequent learning method, which can be Type A, B, and C as described in Arjona-Medina et al. (2019). Type A methods estimate the $Q$-values. They can be estimated directly according to Eq. (2) assuming an optimal redistribution (Type A variant i). $Q$-values can be corrected for a non-optimal reward redistribution by additionally estimating $\kappa$ (Type A variant ii). $Q$-value estimation can use eligibility traces (Type A variant iii). Type B methods use the redistributed rewards for policy gradients like Proximal Policy Optimization (PPO) Schulman et al. (2018). Type C methods use TD learning like $Q$-learning Watkins (1989), where immediate and future reward must be drawn together as typically done. For all these learning methods, demonstrations can be used for initialization (e.g. experience replay buffer) or pre-training (e.g. policy network with behavioral cloning).

**Non-optimal reward redistribution and Align-RUDDER.** According to Theorem 1, non-optimal reward redistributions do not change the optimal policies. The value $\kappa(T-t-1, t)$ measures the remaining delayed reward. The smaller $\kappa$ is, the faster is the learning process. For Monte Carlo (MC) estimates, smaller $\kappa$ reduces the variance of the future rewards, and, therefore the variance of the estimation. For temporal difference (TD) estimates, smaller $\kappa$ reduces the amount of information that has to flow back. Align-RUDDER dramatically reduces the amount of delayed rewards by identifying key events via multiple sequence alignment, to which reward is redistributed. For an episodic MDP, a reward that is redistributed to time $t$ reduces all $\kappa(m, \tau)$ with $t \leqslant \tau < T$ by the expectation of the reward. Therefore, in most cases Align-RUDDER makes $\kappa$-values much smaller.

## 3  Reward Redistribution by Sequence Alignment

In bioinformatics, sequence alignment identifies similarities between biological sequences to determine their evolutionary relationship (Needleman & Wunsch, 1970; Smith & Waterman, 1981). Align-RUDDER uses such alignment techniques to align two or more high return demonstrations. We assume that the demonstrations follow the same underlying strategy, therefore they are similar to each other and can be aligned. However, if there is not an underlying strategy, then the alignment will fail. Events will not receive a high redistributed reward and the reward is given at sequence end, when the redistributed reward is corrected. Then our method does not give an advantage, since the assumption of an underlying strategy is violated. The alignment provides a profile model given as a consensus sequence (the strategy), a frequency matrix, or a Position-Specific Scoring Matrix (PSSM) (Stormo et al., 1982). The redistributed reward of a new sequence is the difference of the scores of consecutive sub-sequences when aligned to the profile model, where the alignment score is the return-decomposition function $g$. This difference indicates how much of the return is gained or lost by a sequence element, which is the main idea of return decomposition: $g((s, a)_{0:t}) - g((s, a)_{0:t-1})$. Align-RUDDER can be cast as following the underlying strategy, which has been extracted by the profile model, therefore is a strategy-matching imitation learning method. The new reward redistribution

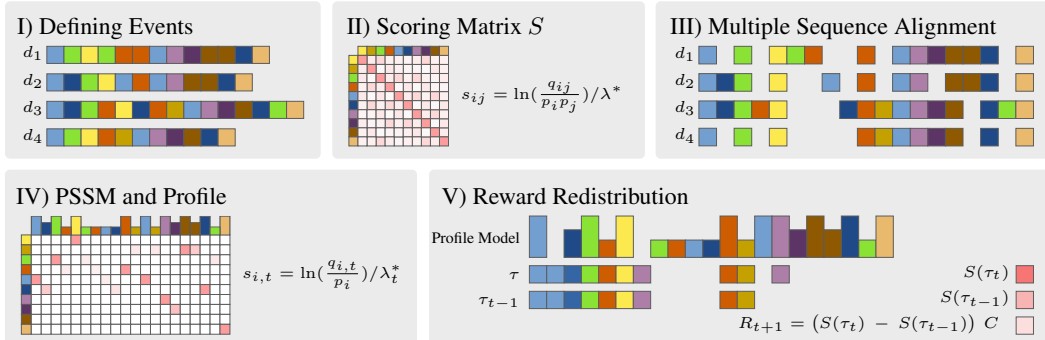

Figure 3: The five steps of Align-RUDDER's reward redistribution.

approach consists of five steps, see Fig. 3: (I) Define events to turn episodes of state-action sequences into sequences of events. (II) Determine an alignment scoring scheme, so that relevant events are aligned to each other. (III) Perform a multiple sequence alignment (MSA) of the demonstrations. (IV) Compute the profile model and the PSSM. (V) Redistribute the reward: Each sub-sequence $\tau_t$ of a new episode $\tau$ is aligned to the profile. The redistributed reward $R_{t+1}$ is proportional to the difference of scores $S$ based on the PSSM given in step (IV), i.e. $R_{t+1} \propto S(\tau_t) - S(\tau_{t-1})$.

**(I) Defining Events.** Events are defined as clusters of differences of consecutive states or state-actions. For clustering, we require representations of state-action pairs. If states do not have a rich representation, we suggest to use the "successor representation" (Dayan, 1993). For a given policy that provides state-action sequences, a successor representation supplies a similarity matrix that indicates the connectivity of two states. As given policy, we use the demonstrations combined with state-action sequences generated by a random policy. Also "successor features" (Barreto et al., 2017) may be used. Subsequently, we use similarity-based clustering like affinity propagation (AP) (Frey & Dueck, 2007). For AP the similarity matrix does not have to be symmetric and the number of clusters not to be known. Each cluster corresponds to an event. Alignment techniques assume sequences composed of few events, e.g. 20 events. If there are too many events, then for two demonstrations only few events can be matched, therefore aligned demonstrations cannot be distinguished from random alignments. This effect is known in bioinformatics as "Inconsistency of Maximum Parsimony" (Felsenstein, 1978). A sequence of events is obtained from a state-action sequence by mapping state-action pairs $(s, a)$ to their cluster identifier $e$ (the event) via a function $f$: $e = f(s, a)$.

**(II) Determining the Alignment Scoring System.** A scoring matrix $\mathbb{S}$ has entries $\mathbb{s}_{i,j}$ that give the score for aligning event $i$ with $j$. The score $S_{\mathrm{MSA}}$ of a MSA is the sum of all pairwise scores: $S_{\mathrm{MSA}} = \sum_{i,j,i<j} \sum_{t=0}^{L} \mathbb{s}_{x_{i,t},x_{j,t}}$, where $x_{i,t}$ means that event $x_{i,t}$ is at position $t$ for sequence $\tau_i = e_{i,0:T}$ in the alignment, analog for $x_{j,t}$ and the sequence $\tau_j = e_{j,0:T}$, and $L$ is the alignment length. Since gaps are present in the alignment, $L \geq T$ and $x_{i,t} \neq e_{i,t}$, . Event $i$ is observed with probability $p_i$ in the demonstrations. An MSA algorithm maximizes the score $S_{\mathrm{MSA}}$ and, thereby, aligns events $i$ and $j$ with probability $q_{ij}$ for demonstrations. According to Theorem 2 and Equation [3] in Karlin & Altschul (1990), asymptotically with the sequence length, we have $\mathbb{s}_{i,j} = \ln(q_{ij}/(p_i p_j))/\lambda^*$, where $\lambda^*$ is the unique positive root of $\sum_{i=1,j=1}^{n,n} p_i p_j \exp(\lambda \mathbb{s}_{i,j}) = 1$. High values of $q_{ij}$ should indicate relevant events of the strategy. For defining $\mathbb{s}_{i,j}$ by above formula, we therefore set $q_{ij} = p_i - \epsilon$ for $i = j$ and $q_{ij} = \epsilon/(n-1)$ for $i \neq j$, where $n$ is the number of events. More details in appendix Sec. A.2.

**(III) Multiple sequence alignment (MSA).** MSA produces a guiding tree (agglomerative hierarchical clustering) from pairwise alignments between all demonstrations, where demonstrations with the same strategy are in the same cluster. Each cluster is aligned separately via MSA to construct different strategies. However, if demonstrations do not cluster, the alignment may fail. ClustalW (Thompson et al., 1994) can be used for MSA. More details in appendix Sec. A.2.

**(IV) Position-Specific Scoring Matrix (PSSM) and Profile.** From the final alignment, we construct a) an MSA profile (column-wise event frequencies $q_{i,j}$) and b) a PSSM (Stormo et al., 1982) which is used for aligning new sequences to the profile of the MSA. More details in appendix Sec. A.2.

**(V) Reward Redistribution.** The reward redistribution is based on the profile model. A sequence $\tau = e_{0:T}$ ($e_t$ is event at position $t$) is aligned to the profile, which gives the score $S(\tau) = \sum_{t=0}^{L} \mathbb{s}_{x_t,t}$. Here, $\mathbb{s}_{i,t}$ is the alignment score for event $i$ and $x_t$ is the event of $\tau$ at position $t$ in the alignment. $L$ is the profile length, where $L \geq T$ and $x_t \neq e_t$, because of gaps in the alignment. If $\tau_t = e_{0:t}$ is the prefix sequence of $\tau$ of length $t + 1$, then the reward redistribution $R_{t+1}$ for $0 \leqslant t \leqslant T$ is

$$R_{t+1} \;=\; (S(\tau_t) \;-\; S(\tau_{t-1}))\, C \;=\; g((s,a)_{0:t}) - g((s,a)_{0:t-1}), \; R_{T+2} \;=\; \tilde{G}_0 - \sum_{t=0}^{T} R_{t+1}, \quad (3)$$

where $C = \mathrm{E}_{\mathrm{demo}}\left[\tilde{G}_0\right] \,/\, \mathrm{E}_{\mathrm{demo}}\left[\sum_{t=0}^{T} S(\tau_t) - S(\tau_{t-1})\right]$ and $\tilde{G}_0 = \sum_{t=0}^{T} \tilde{R}_{t+1}$ is the original return of the sequence $\tau$ and $S(\tau_{-1}) = 0$. $\mathrm{E}_{\mathrm{demo}}$ is the expectation over demonstrations, and $C$ scales $R_{t+1}$ to the range of $\tilde{G}_0$. $R_{T+2}$ is the correction of the redistributed reward (Arjona-Medina et al., 2019), with zero expectation for demonstrations: $\mathrm{E}_{\mathrm{demo}}\left[R_{T+2}\right] = 0$. Since $\tau_t = e_{0:t}$ and $e_t = f(s_t, a_t)$, we can set $g((s,a)_{0:t}) = S(\tau_t)C$. We ensure strict return equivalence by $G_0 = \sum_{t=0}^{T+1} R_{t+1} = \tilde{G}_0$. The redistributed reward depends only on the past: $R_{t+1} = h((s,a)_{0:t})$.
*Sub-tasks.* The reward redistribution identifies sub-tasks as alignment positions with high redistributed reward. It also determines the terminal states and assigns reward for solving the sub-tasks. However, reward redistribution and Align-RUDDER cannot guarantee that the reward is Markov. For redistributed Markov reward, options (Sutton et al., 1999), MAXQ (Dietterich, 2000), or recursive option composition (Silver & Ciosek, 2012) can be used for hierarchical reinforcement learning.

## 4    EXPERIMENTS

Align-RUDDER is compared on two artificial tasks with sparse & delayed rewards and few demonstrations to Behavioral Cloning with $Q$-learning (BC+$Q$), Soft $Q$ Imitation Learning (SQIL) (Reddy et al., 2020), and Deep $Q$-learning from Demonstrations (DQfD) (Hester et al., 2018). GAIL (Ho & Ermon, 2016), failed to solve the two artificial tasks, as reported previously for similar tasks (Reddy et al., 2020). Then, we test Align-RUDDER on the complex MineCraft `ObtainDiamond` task with episodic rewards (Guss et al., 2019b). All experiments use finite time horizon MDPs with $\gamma = 1$ and episodic reward. More details in appendix Sec. A.4.

**Artificial tasks (I) and (II).** They are variations of the gridworld *rooms example* (Sutton et al., 1999), where cells (locations) are the MDP states. In our setting, the states do not have to be time-aware for ensuring stationary optimal policies but the unobserved used-up time introduces a random effect. The grid is divided into rooms. The agent's goal is to reach a target from an initial state with fewest steps. It has to cross different rooms, which are connected by doors, except for the first room, which is only connected to the second room by a *portal*. If the agent is at the portal entry cell of the first room then it is teleported to a fixed portal arrival cell in the second room. The location of the portal entry cell is randomly chosen for each episode, while the portal arrival cell is fixed across episodes. The portal entry cell location is given in the state for the first room. The portal is introduced to avoid that BC initialization alone solves the task. It enforces that going to the portal entry cells is learned, when they are at positions not observed in demonstrations. At every location, the agent can move *up, down, left, right* if it stays on the grid. The state transitions are stochastic, except for teleportation. An episode ends after $T = 200$ time steps. If the agent arrives at the target, then at the next step it goes into an absorbing state where it stays until $T = 200$ without receiving further rewards. Reward is only given at the end of the episode. The final reward is 3 when reaching the target, and 1 otherwise. To enforce that the agent reaches the goal with the fewest steps possible, a small negative reward of $-0.01$ is given at every time step. Demonstrations are generated by an optimal policy with 0.2 exploration rate.

The five steps of Align-RUDDER's reward redistribution are: (1) Events are clusters of states obtained by Affinity Propagation using as similarity the successor representation of the states based on demonstrations. (2) The scoring matrix is obtained according to (II), using $\epsilon = 0$ and setting all off-diagonal values of the scoring matrix to $-1$. (3) ClustalW is used for the MSA of the demonstrations with zero gap penalties and no biological options. (4) The MSA supplies a profile model and a PSSM as in (IV). (5) Sequences generated by the agent are mapped to sequences of events according to (I). Reward is redistributed via differences of profile alignment scores of consecutive sub-sequences according to Eq. (3) using the PSSM. **Sub-tasks.** The reward redistribution determines sub-tasks like

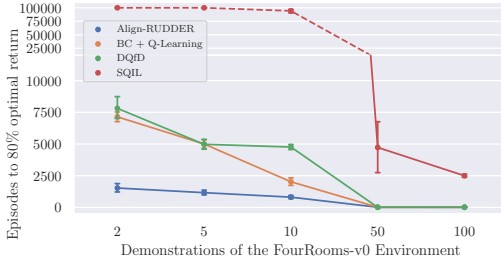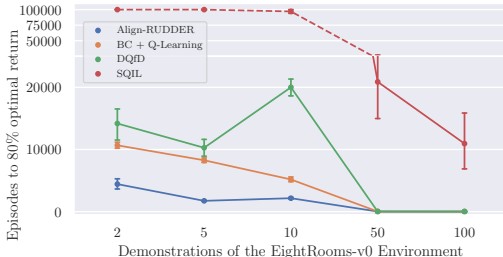

Figure 4: Comparison of Align-RUDDER and other methods on Task (I) (left) and Task (II) (right) with respect to the number of episodes required for learning on different numbers of demonstrations. Results are the average over 100 trials. Align-RUDDER significantly outperforms all other methods.

doors or portal arrival. The sub-tasks partition the $Q$-table into sub-tables that represent a sub-agent. In the tabular case, defining sub-tasks has no effect on learning if compared to a single $Q$-table.

All compared methods learn a $Q$-table and use an $\epsilon$-greedy policy with $\epsilon = 0.2$. The $Q$-table is initialized by behavioral cloning (BC). The state-action pairs which are not initialized since they are not visited in the demonstrations get an initialization by drawing a sample from a normal distribution with mean $1$ and standard deviation $0.5$ (avoiding equal $Q$-values). Align-RUDDER learns the $Q$-table via RUDDER's $Q$-value estimation with $\kappa$ correction (Type A variant ii from above). For BC+$Q$, SQIL, and DQfD a $Q$-table is learned by $Q$-learning. Hyperparameters are selected via grid search using the same amount of time for each method. For different numbers of demonstrations, performance is measured by the number of episodes to achieve 80% of the average return of the demonstrations. A Wilcoxon rank-sum test determines the significance of performance differences between Align-RUDDER and the other methods.

**Task (I)** environment is a $12 \times 12$ gridworld with four rooms. The target is in room #4 and the start is in room #1 with 20 portal entry locations. The state contains the portal entry for each episode. Fig. 4 shows the number of episodes required for achieving 80% of the average reward of the demonstrations for different numbers of demonstrations. Results are averaged over 100 trials. **Align-RUDDER significantly outperforms all other methods, for $\leqslant 10$ demonstrations ($p$-values $< 10^{-10}$).**
**Task (II)** is a $12 \times 24$ gridworld with eight rooms: target in room #8, and start in room #1 with 20 portal entry locations. Fig. 4 shows the results with settings as in Task (I). **Align-RUDDER significantly outperforms all other methods, for $\leqslant 10$ demonstrations ($p$-values $< 10^{-19}$).** More details and 7 additional experiments with different stochasticity values are in the appendix Sec. A.4.

**MineCraft.** We further test Align-RUDDER on the challenging MineCraft `ObtainDiamond` task from the MineRL dataset (Guss et al., 2019b). We do not use the intermediate rewards given by achieving sub-goals from the challenge, since Align-RUDDER is supposed to discover such sub-goals automatically via reward redistribution. We only give reward for mining the diamond. This requires resource gathering and tool building in a hierarchical way. To the best of our knowledge, no pure learning method (sub-goals are also learned) has mined a diamond yet (Scheller et al., 2020). The dataset contains demonstrations from human players. However, the number of demonstrations is insufficient to directly learn a single policy from them (117 demonstrations, 67 mined a diamond).

Implementation: (1) A state consists of a visual input and an inventory (incl. equip state). Both inputs are normalized to the same information, that is, the same number of components and the same variance. We cluster the differences of consecutive states (Arjona-Medina et al., 2019). Very large clusters are removed and small merged giving 19 clusters corresponding to events, which are characterized by inventory changes. Finally, demonstrations are mapped to sequences of events. (2) The scoring matrix is computed according to (II). (3) The 10 shortest demonstrations that obtained a diamond are aligned by ClustalW with zero gap penalties and no biological options. (4) The multiple alignment gives a profile model and a PSSM. (5) Reward is redistributed via differences of profile alignment scores of consecutive sub-sequences according to Eq. (3) using the PSSM. Based on the reward redistribution we define sub-goals. Sub-goals are identified as profile model positions which obtain an average redistributed reward above a threshold for the demonstrations. Demonstration sub-sequences between sub-goals are considered as demonstrations for the sub-tasks. New sub-sequences

| Method | Team Name | HRL | Sup | Online | Automated | 🟫 | 🟫 | ⛏ | ⬛ | ⛏ | ⬛ | ⬜ | ⛏ | 💎 |
|--------|-----------|-----|-----|--------|-----------|---|---|---|---|---|---|---|---|---|
| Align-RUDDER | Ours | ✓ | ✓ | ✗ | ✓ | | | | | | | | | |
| DQfD | CDS | ✓ | ✗ | ✓ | ✗ | | | | | | | | | |
| BC | MC_RL | ✓ | ✓ | ✗ | — | | | | | | | | | |
| CLEAR | I4DS | ✗ | ✓ | ✓ | ✓ | | | | | | | | | |
| Options&PPO | CraftRL | ✓ | ✓ | ✗ | ✗ | | | | | | | | | |
| BC | UEFDRL | ✗ | ✓ | ✗ | ✓ | | | | | | | | | |
| SAC | TD240 | ✗ | ✓ | ✓ | ✓ | | | | | | | | | |
| MLSH | LAIR | ✓ | ✓ | ✗ | ✓ | | | | | | | | | |
| Rainbow | Elytra | ✗ | ✓ | ✓ | ✓ | | | | | | | | | |
| PPO | karolisram | ✗ | ✓ | ✗ | ✓ | | | | | | | | | |

Table 1: Maximum item score of methods on the MineCraft task. Results are from (Milani et al., 2020; Skrynnik et al., 2019; Kanervisto et al., 2020; Scheller et al., 2020). "Automated": Sub-goals/sub-tasks are found automatically. Demonstrations are used for hierarchical reinforcement learning ("HRL"), supervised training ("Sup"), with online data ("Online"). Methods: Soft-Actor Critic (SAC, Haarnoja et al. (2018)), DQfD, Meta Learning Shared Hierarchies (MLSH, Frans et al. (2018)), Rainbow (Hessel et al., 2017), PPO, and BC. Align-RUDDER may have advantages as it did not participate at the challenge. Though the challenge rewards that hint at sub-tasks are not used.

generated by the agent are aligned to the profile model to determine whether a sub-goal is achieved. The redistributed reward between two sub-goals is given at the end of the sub-sequence, therefore, the sub-tasks have also episodic reward. Fig. A.11 in the appendix shows how sub-goals are identified. Sub-agents are pre-trained on the demonstrations for the sub-tasks using BC, and further trained in the environment using Proximal Policy Optimization (Schulman et al., 2018).

Our main agent can perform all actions but additionally can execute sub-agents and learns via the redistributed reward (return-equivalent MDP). The main agent corresponds to and is treated like a Manager module (Vezhnevets et al., 2017). The main agent is initialized by executing sub-agents according to the alignment but can deviate from this strategy. More implementation details can be found in the appendix. Using only 10 demonstrations, Align-RUDDER is able to learn to mine a diamond. A diamond is obtained in 0.1% of the cases. With 0.5 success probability for each of the 31 extracted sub-tasks (skilled agents), the resulting success rate for mining the diamond would be $4.66 \times 10^{-10}$. Tab. 1 shows a comparison of methods on the MineCraft MineRL dataset by the maximum item score (Milani et al., 2020). Results are taken from (Milani et al., 2020), in particular from Figure 2, and completed by (Skrynnik et al., 2019; Kanervisto et al., 2020; Scheller et al., 2020). Align-RUDDER was not evaluated during the challenge, therefore may have advantages. Though it did not receive the intermediate rewards provided by the challenge that hint at sub-tasks.

For each agent and its sub-task, we estimate the success rate and its improvement during fine-tuning by averaging over return of multiple runs (see Fig. 5).

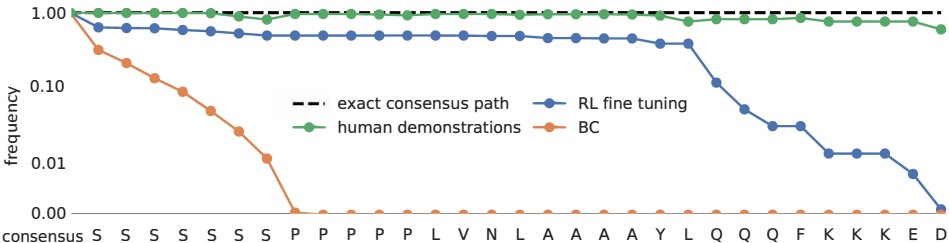

Figure 5: Comparing the consensus frequencies between behavioral cloning (BC, orange), where fine-tuning starts, the fine-tuned model (blue), and human demonstrations (green).

**Conclusions.** We have introduced Align-RUDDER to solve highly complex tasks with delayed and sparse reward from few demonstrations. On the MineCraft `ObtainDiamond` task, Align-RUDDER is, to the best of our knowledge, the first pure learning method to mine a diamond.

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

# A APPENDIX

## CONTENTS OF THE APPENDIX

## LIST OF FIGURES

### A.1 INTRODUCTION TO THE APPENDIX

This is the appendix to the paper "Align-RUDDER: Learning from few Demonstrations by Reward Redistribution". The appendix aims at supporting the main document and provides more detailed information about the implementation of our method for different tasks. The content of this document is summarized as follows:

• Section A.2 describes the five steps of Align-RUDDER's reward redistribution in more detail. In particular, the scoring systems are described in more detail. • Section A.3 provides a brief overview of sequence alignment methods and the hyperparameters used in our experiments. • Section A.4 provides figures and tables to support the results of the experiments in Artificial Tasks (I) and (II). • Section A.5 explains in detail the experiments conducted in the Minecraft *ObtainDiamond* task.

### A.2 THE FIVE STEPS OF ALIGN-RUDDER'S REWARD REDISTRIBUTION

The new reward redistribution approach consists of five steps, see Fig. A.1: (I) Define events to turn episodes of state-action sequences into sequences of events. (II) Determine an alignment scoring scheme, so that relevant events are aligned to each other. (III) Perform a multiple sequence alignment (MSA) of the demonstrations. (IV) Compute the profile model and the PSSM. (V) Redistribute the reward: Each sub-sequence $\tau_t$ of a new episode $\tau$ is aligned to the profile. The redistributed reward $R_{t+1}$ is proportional to the difference of scores $S$ based on the PSSM given in step (IV), i.e. $R_{t+1} \propto S(\tau_t) - S(\tau_{t-1})$.

**(I) Defining Events.** Alignment techniques assume that sequences consist of few symbols, e.g. about 20 symbols, the events. It is crucial to keep the number of events small in order to increase

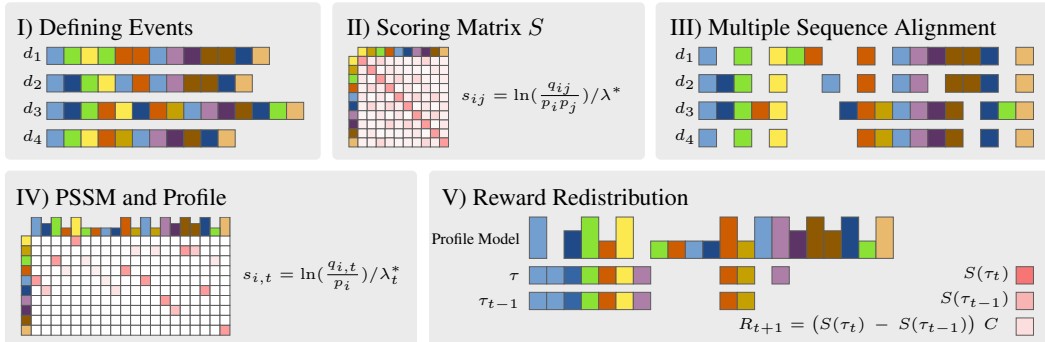

Figure A.1: The five steps of Align-RUDDER's reward redistribution.

the difference between a random alignment and an alignment of demonstrations. If there are many events, then two demonstrations might have few events that can be matched, which cannot be well distinguished from random alignments. This effect is known in bioinformatics as "Inconsistency of Maximum Parsimony" (Felsenstein, 1978). The events can be the original state-action pairs, clusters thereof, or other representations of state-action pairs, e.g. indicating changes of inventory, health, energy, skills etc. In general, we define events as a cluster of states or state-actions. A sequence of events is obtained from a state-action sequence by substituting states or state-actions by their cluster identifier. In order to cluster states, a similarity measure between them is required. We suggest to use the "successor representation" (Dayan, 1993) of the states, which gives a similarity matrix based on how connected two states are given a policy. Successor representation have been used before (Machado et al., 2017; Ramesh et al., 2019) to obtain important events, for option learning. For computing the successor representation, we use the demonstrations combined with state-action sequences generated by a random policy. For high dimensional state spaces "successor features" (Barreto et al., 2017) can be used. We use similarity-based clustering methods like affinity propagation (AP) (Frey & Dueck, 2007). For AP the similarity matrix does not have to be symmetric and the number of clusters need not be known. State action pairs $(s, a)$ are mapped to events $e$ by a function $f$: $e = f(s, a)$.

**(II) Determining the Alignment Scoring System.** Alignment algorithms distinguish similar sequences from dissimilar sequences using a scoring system. A scoring matrix $\mathbb{S}$ has entries $\mathbb{s}_{i,j}$ that give the score for aligning event $i$ with $j$. The MSA score $S_{\text{MSA}}$ of a multiple sequence alignment is the sum of all pairwise scores: $S_{\text{MSA}} = \sum_{i,j,i<j} \sum_{t=0}^{L} \mathbb{s}_{x_{i,t}, x_{j,t}}$, where $x_{i,t}$ means that event $x_{i,t}$ is at position $t$ for sequence $\tau_i = e_{i,0:T}$ in the alignment, analog for $x_{j,t}$ and the sequence $\tau_j = e_{j,0:T}$, and $L$ is the alignment length. Note that $L \geq T$ and $x_{i,t} \neq e_{i,t}$, since gaps are present in the alignment. In the alignment, events should have the same probability of being aligned as they would have if we know the strategy and align demonstrations accordingly. The theory of high scoring segments gives a scoring scheme with these alignment probabilities (Karlin & Altschul, 1990; Karlin et al., 1990; Altschul et al., 1990). Event $i$ is observed with probability $p_i$ in the demonstrations, therefore a random alignment aligns event $i$ with $j$ with probability $p_i p_j$. An alignment algorithm maximizes the MSA score $S_{\text{MSA}}$ and, thereby, aligns events $i$ and $j$ with probability $q_{ij}$ for demonstrations. High values of $q_{ij}$ means that the MSA often aligns events $i$ and $j$ in the demonstrations using the scoring matrix $\mathbb{S}$ with entries $\mathbb{s}_{i,j}$. According to Theorem 2 and Equation [3] in Karlin & Altschul (1990), asymptotically with the sequence length, we have $\mathbb{s}_{i,j} = \ln(q_{ij}/(p_i p_j))/\lambda^*$, where $\lambda^*$ is the unique positive root of $\sum_{i=1,j=1}^{n,n} p_i p_j \exp(\lambda \mathbb{s}_{i,j}) = 1$ (Equation [4] in Karlin & Altschul (1990)).

We can now choose a desired probability $q_{ij}$ and then compute the scoring matrix $\mathbb{S}$ with entries $\mathbb{s}_{i,j}$. High values of $q_{ij}$ should indicate relevant events for the strategy. A priori, we only know that a relevant event should be aligned to itself, while we do not know which events are relevant. Therefore we set $q_{ij}$ to large values for every $i = j$ and to low values for $i \neq j$. Concretely, we set $q_{ij} = p_i - \epsilon$ for $i = j$ and $q_{ij} = \epsilon/(n-1)$ for $i \neq j$, where $n$ is the number of different possible events. Events with smaller $p_i$ receive a higher score $\mathbb{s}_{i,i}$ when aligned to themselves since this self-match is less often observed when randomly matching events ($p_i p_i$ is the probability of a random self-match). Any prior knowledge about events should be incorporated into $q_{ij}$.

**(III) Multiple sequence alignment (MSA).** MSA first produces pairwise alignments between all demonstrations. Then, a guiding tree (agglomerative hierarchical clustering) is produced via hierarchical clustering sequences, according to their pairwise alignment scores. Demonstrations which follow the same strategy appear in the same cluster in the guiding tree. Each cluster is aligned separately via MSA to address different strategies. However, if there is not a cluster of demonstrations, then the alignment will fail. MSA methods like ClustalW (Thompson et al., 1994) or MUSCLE (Edgar, 2004) can be used.

**(IV) Position-Specific Scoring Matrix (PSSM) and Profile.** From the final alignment, we construct a) an MSA profile (column-wise event frequencies $q_{i,j}$) and b) a PSSM (Stormo et al., 1982) which is used for aligning new sequences to the profile of the MSA. To compute the PSSM (column-wise scores $\mathbb{s}_{i,t}$), we apply Theorem 2 and Equation [3] in Karlin & Altschul (1990). Event $i$ is observed with probability $p_i$ in the data. For each position $t$ in the alignment, we compute $q_{i,t}$, which indicates the frequency of event $i$ at position $t$. The PSSM is $\mathbb{s}_{i,t} = \ln(q_{i,t}/p_i)/\lambda_t^*$, where $\lambda_t^*$ is the single unique positive root of $\sum_{i=1}^{n} p_i \exp(\lambda \mathbb{s}_{i,t}) = 1$ (Equation [1] in Karlin & Altschul (1990)). If we align a new sequence that follows the underlying strategy (a new demonstration) to the profile model, we would see that event $i$ is aligned to position $t$ in the profile with probability $q_{i,t}$.

**(V) Reward Redistribution.** The reward redistribution is based on the profile model. A sequence $\tau = e_{0:T}$ ($e_t$ is event at position $t$) is aligned to the profile, which gives the score $S(\tau) = \sum_{t=0}^{L} \mathbb{s}_{x_t,t}$. Here, $\mathbb{s}_{i,t}$ is the alignment score for event $i$ and $x_t$ is the event of $\tau$ at position $t$ in the alignment. $L$ is the profile length, where $L \geq T$ and $x_t \neq e_t$, because of gaps in the alignment. If $\tau_t = e_{0:t}$ is the prefix sequence of $\tau$ of length $t+1$, then the reward redistribution $R_{t+1}$ for $0 \leqslant t \leqslant T$ is

$$R_{t+1} = (S(\tau_t) - S(\tau_{t-1}))C = g((s,a)_{0:t}) - g((s,a)_{0:t-1}), \ R_{T+2} = \tilde{G}_0 - \sum_{t=0}^{T} R_{t+1}, \ (4)$$

where $C = \mathrm{E}_{\text{demo}}\left[\tilde{G}_0\right] \ / \ \mathrm{E}_{\text{demo}}\left[\sum_{t=0}^{T} S(\tau_t) - S(\tau_{t-1})\right]$ and $\tilde{G}_0 = \sum_{t=0}^{T} \tilde{R}_{t+1}$ is the original return of the sequence $\tau$ and $S(\tau_{-1}) = 0$. $\mathrm{E}_{\text{demo}}$ is the expectation over demonstrations, and $C$ scales $R_{t+1}$ to the range of $\tilde{G}_0$. $R_{T+2}$ is the correction of the redistributed reward (Arjona-Medina et al., 2019), with zero expectation for demonstrations: $\mathrm{E}_{\text{demo}}[R_{T+2}] = 0$. Since $\tau_t = e_{0:t}$ and $e_t = f(s_t, a_t)$, we can set $g((s,a)_{0:t}) = S(\tau_t)C$. We ensure strict return equivalence, since $G_0 = \sum_{t=0}^{T+1} R_{t+1} = \tilde{G}_0$. The redistributed reward depends only on the past, that is, $R_{t+1} = h((s,a)_{0:t})$. For computational efficiency, the alignment of $\tau_{t-1}$ can be extended to one for $\tau_t$, like exact matches are extended to high-scoring sequence pairs with the BLAST algorithm (Altschul et al., 1990; 1997).

*Sub-tasks.* The reward redistribution identifies sub-tasks, which are alignment positions with high redistributed reward. It also determines the terminal states and automatically assigns reward for solving the sub-tasks. However, reward redistribution and Align-RUDDER cannot guarantee that the reward is Markov. For redistributed reward that is Markov, the option framework (Sutton et al., 1999), the MAXQ framework (Dietterich, 2000), or recursive composition of option models (Silver & Ciosek, 2012) can be used as subsequent approaches to hierarchical reinforcement learning.

## A.3 SEQUENCE ALIGNMENT

In bioinformatics, sequence alignment identifies regions of significant similarity among different biological sequences to establish evolutionary relationships between those sequences. In 1970, Needleman and Wunsch proposed a global alignment method based on dynamic programming (Needleman & Wunsch, 1970). This approach ensures the best possible alignment given a substitution matrix, such as PAM (Dayhoff, 1978) or BLOSUM (Henikoff & Henikoff, 1992), and other parameters to penalize gaps in the alignment. The method of Needlemann and Wunsch is of $O(mn)$ complexity both in memory and time, which could be prohibitive in long sequences like genomes. An optimization of this method by Hirschberg (1975), reduces memory to $O(m+n)$, but still requires $O(mn)$ time.

Later, Smith and Waterman developed a local alignment method for sequences (Smith & Waterman, 1981). It is a variation of Needleman and Wunsch's method, keeping the substitution matrix and the gap-scoring scheme but setting cells in the similarity matrix with negative scores to zero. The complexity for this algorithm is of $O(n^2M)$. Osamu Gotoh published an optimization of this method, running in $O(mn)$ runtime (Gotoh, 1982).

The main difference between both methods is the following:

- The global alignment method by Needleman and Wunsch aligns the sequences fixing the first and the last position of both sequences. It attempts to align every symbol in the sequence, allowing some gaps, but the main purpose is to get a global alignment. This is especially useful when the two sequences are highly similar. For instance:

```
ATCGGATCGACTGGCTAGATCATCGCTGG
CGAGCATC-ACTGTCT-GATCGACCTTAG
*  *** **** ** ****   *   * *
```

- As an alternative to global methods, the local method of Smith and Waterman aligns the sequences with a higher degree of freedom, allowing the alignment to start or end with gaps. This is extremely useful when the two sequences are substantially dissimilar in general but suspected of having a highly related sub region.

```
ATCAAGGAGATCATCGCTGGACTGAGTGGCT----ACGTGGTATGT
ATC----CGATCATCGCTGG-CTGATCGACCTTCTACGT-------
***      ************ ****   * *      ****
```

**A.3.0.1  Multiple Sequence Alignment algorithms.**  The sequence alignment algorithms by Needleman and Wunsch and Smith and Waterman are limited to aligning two sequences. The approaches for generalizing these algorithms to multiple sequences can be classified into four categories:

- Exact methods (Wang & Jiang, 1994).
- Progressive methods: ClustalW (Thompson et al., 1994), Clustal Omega (Sievers et al., 2014), T-Coffee (Notredame et al., 2000).
- Iterative and search algorithms: DIALIGN (Morgenstern, 2004), MultiAlign (Corpet, 1988).
- Local methods: eMOTIF (Mccammon & Wolynes, 1998), PROSITE (Bairoch & Bucher, 1994).

For more details, visit *Sequence Comparison: Theory and methods* (Chao & Zhang, 2009).

In our experiments, we use ClustalW from Biopython (Cock et al., 2009) with the following parameters:

```
clustalw2 -ALIGN -CLUSTERING=UPGMA -NEGATIVE " \
        "-INFILE={infile} -OUTFILE={outfile} " \
        "-PWMATRIX={scores} -PWGAPOPEN=0 -PWGAPEXT=0 " \
        "-MATRIX={scores} -GAPOPEN=0 -GAPEXT=0 -CASE=UPPER " \
        "-NOPGAP -NOHGAP -MAXDIV=0 -ENDGAPS -NOVGAP " \
        "-NEWTREE={outputtree} -TYPE=PROTEIN -OUTPUT=GDE
```

where the `PWMATRIX` and `MATRIX` are computed according to step (II) in Sec. 3 of the main paper.

## A.4 ARTIFICIAL TASK EXPERIMENTS

This section provides additional information that supports the results reported in the main paper for Artificial Tasks (I) and (II).

### A.4.1 HYPERPARAMETER SELECTION

For (BC)+$Q$-Learning and Align-RUDDER, we performed a grid search to select the learning rate from the following values $[0.1, 0.05, 0.01]$. We used 20 different seeds for each value and each number of demonstrations and then selected the setting with the highest success for all number of demonstrations. The final learning rate for (BC)+$Q$-Learning and DQfD is $0.01$ and for Align-RUDDER it is $0.1$.

For DQfD, we set the experience buffer size to $30,000$ and the number of experiences sampled at every timestep to $10$. The DQfD loss weights are set to $0.01$, $0.01$ and $1.0$ for the $Q$-learning loss term, $n$-step loss term and the expert loss respectively during pre-training. During online learning, we change the loss terms to $1.0$, $1.0$ and $0.01$ for the $Q$-learning loss term, $n$-step loss term and the expert loss term. This was necessary to enable faster learning for DQfD. The expert action margin is $0.8$.

For successor representation, we use a learning rate of $0.1$ and a gamma of $0.99$. We update the successor table multiple times using the same transition (state, action, next state) from the demonstration.

For affinity propagation, we use a damping factor of $0.5$ and set the maximum number of iterations to $1000$. Furthermore, if we obtain more than 15 clusters, then we combine clusters based on the similarity of the cluster centers.

### A.4.2 FIGURES

Figure A.5 shows sample trajectories in the FourRooms and EightRooms environment, with the initial and target positions marked in red and green respectively. Figure A.2 shows the clusters after performing clustering with Affinity Propagation using the successor representation with 25 demonstrations and an environment with 1% stochasticity on the transitions. Different colors indicate different clusters. Figures A.3 and A.4 show clusters for different environment settings. Figure A.3 shows clusters when using 10 demonstrations and for Figure A.4 environments with 5% stochasticity on transitions was used. Figure A.6 shows the reward redistribution for the given example trajectories in the FourRooms and EightRooms environments.

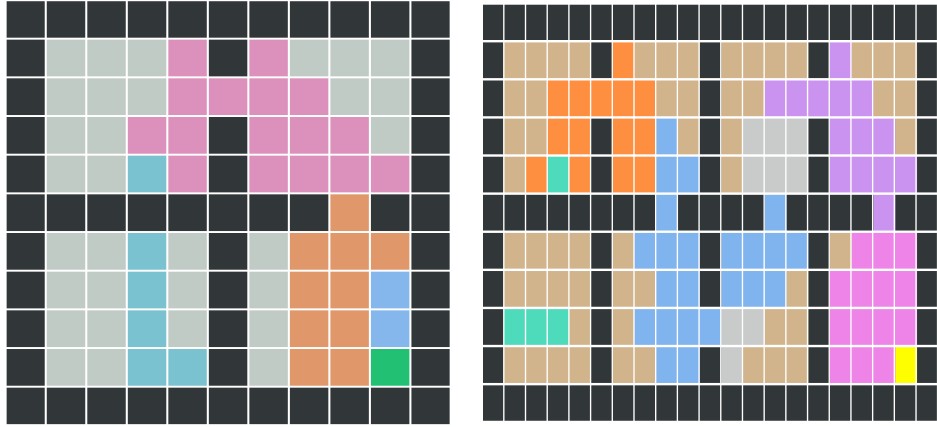

Figure A.2: Examples of clusters formed in the FourRooms (left) and EightRooms (right) environment with 1% stochasticity on the transitions after performing clustering with Affinity Propagation using the successor representation with 25 demonstrations. Different colors represent different clusters.

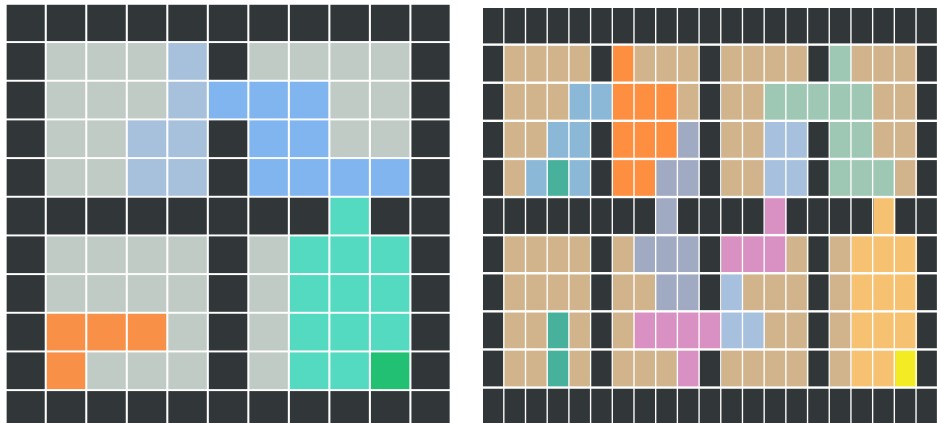

Figure A.3: Examples of clusters formed in the FourRooms (left) and EightRooms (right) environment with 1% stochasticity on the transitions after performing clustering with Affinity Propagation using the successor representation with 10 demonstrations. Different colors represent different clusters.

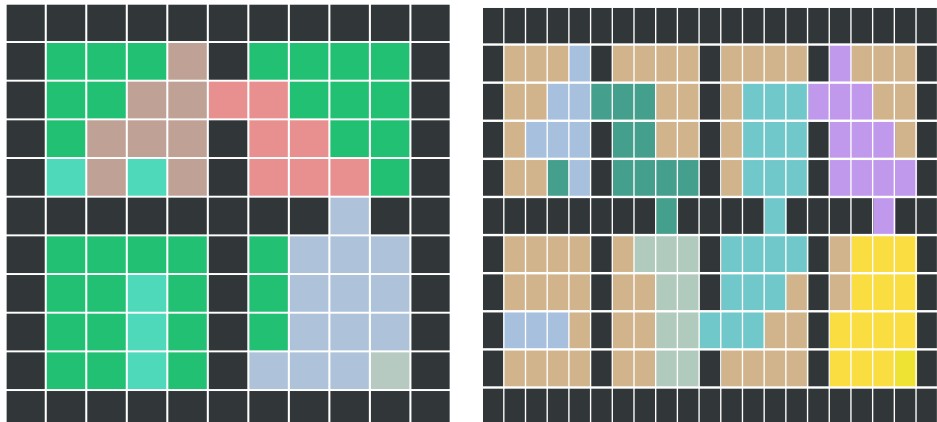

Figure A.4: Examples of clusters formed in the FourRooms (left) and EightRooms (right) environment with 5% stochasticity on the transitions after performing clustering with Affinity Propagation using the successor representation with 25 demonstrations. Different colors represent different clusters.

### A.4.3 ARTIFICIAL TASK p-VALUES

Tables A.1 and A.2 show the $p$-values obtained by performing a Mann-Whitney-U test between Align-RUDDER and BC+$Q$-Learning and DQfD respectively.

| | 2 | 5 | 10 | 50 | 100 |
|---|---|---|---|---|---|
| Align-RUDDER vs. BC+$Q$-Learn. | 8.8361e-31 | 2.8198e-30 | 1.1380e-09 | 0.3547 | 0.1558 |
| Align-RUDDER vs. DQfD | 2.6984e-29 | 4.3348e-30 | 1.3456e-32 | 1.0000 | 0.9999 |

Table A.1: $p$-values for Artificial Task (I), FourRooms, obtained by performing a Mann-Whitney-U test.

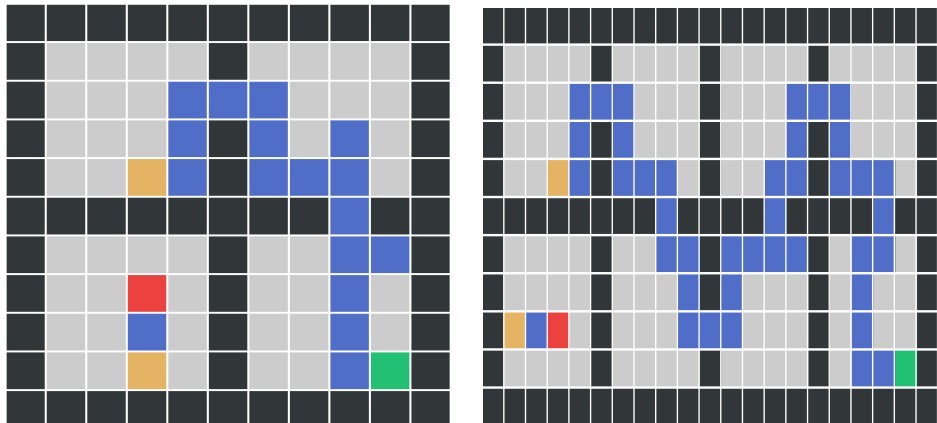

Figure A.5: Exemplary trajectories in the FourRooms (left) and EightRooms (right) environments. Initial position is indicated in red, the portal between the first and second room in yellow and the goal in green.

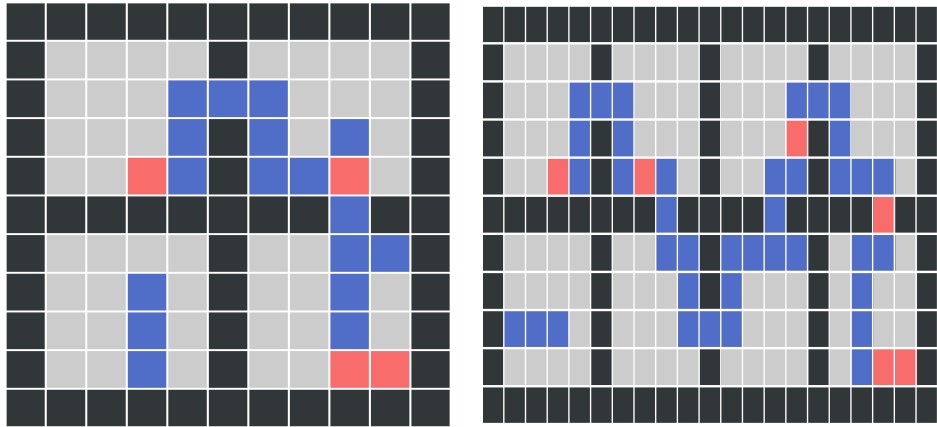

Figure A.6: Reward redistribution for the above trajectories in the FourRooms (left) and EightRooms (right) environments.

### A.4.4 STOCHASTIC ENVIRONMENTS

Figure A.7 shows results for the FourRooms environment with different levels of stochasticity (5%, 10%, 15%, 25% and 40%) on the transitions. Figure A.8 shows results for the EightRooms environment with different levels of stochasticity (5% and 10%) on the transitions.

| | 2 | 5 | 10 | 50 | 100 |
|---|---|---|---|---|---|
| Align-RUDDER vs. BC+$Q$-Learn. | 4.5409e-20 | 1.2776e-34 | 4.8883e-25 | 0.3730 | 0.6096 |
| Align-RUDDER vs. DQfD | 1.2202e-08 | 8.9356e-20 | 5.6255e-31 | 0.9999 | 0.9964 |

Table A.2: $p$-values for Artificial Task (II), EightRooms, obtained by performing a Mann-Whitney-U test.

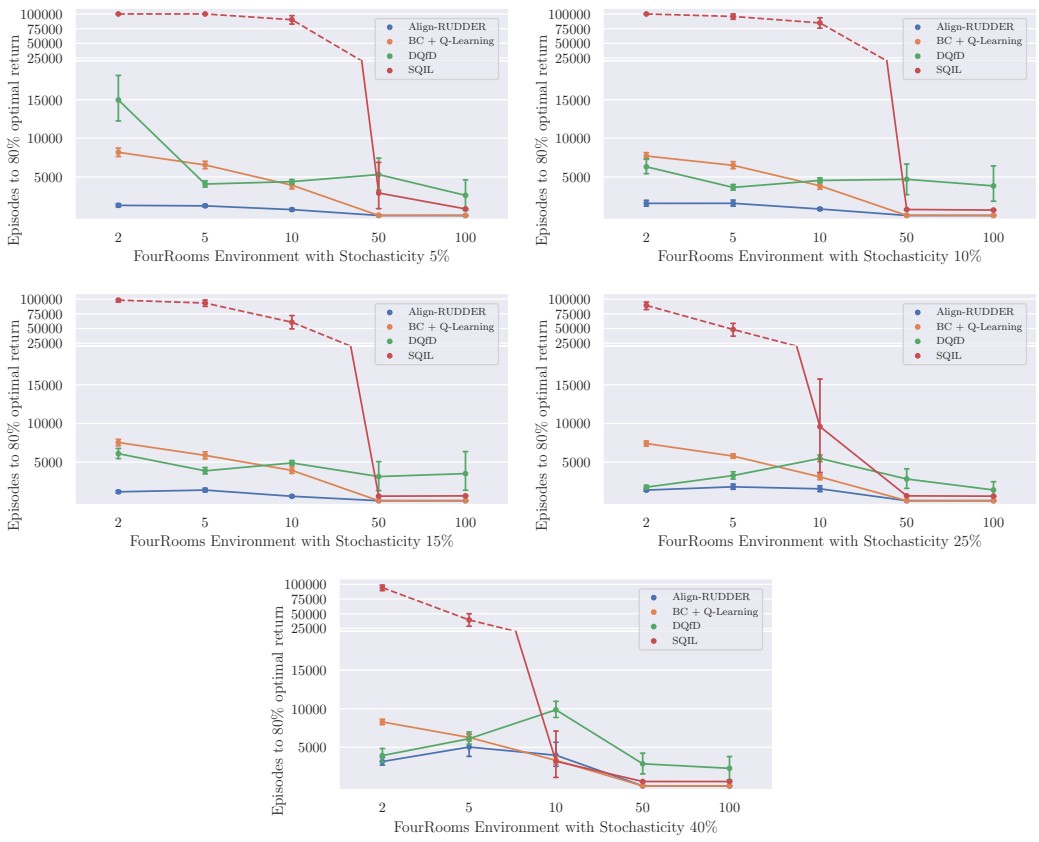

Figure A.7: Comparison of Align-RUDDER and other methods on Task (I) (FourRooms) with increasing levels of stochasticity (from top left to bottom: 5%, 10%, 15%, 25% and 40%). Results are the average over 50 trials.

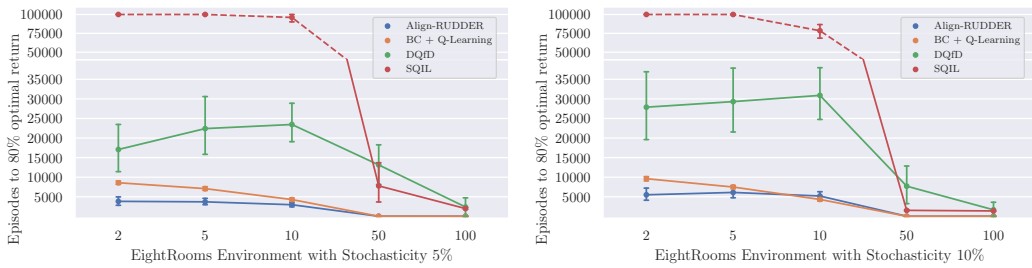

Figure A.8: Comparison of Align-RUDDER and other methods on Task (II) (EightRooms) with increasing levels of stochasticity (from top left to bottom: 5%, 10%). Results are the average over 50 trials.

## A.5 MINECRAFT EXPERIMENTS

In this section we explain in detail the implementation of Align-RUDDER for solving the task *ObtainDiamond*.

### A.5.1 MINECRAFT

We show that our approach can be applied to complex tasks by evaluating it on the MineRL Minecraft dataset (Guss et al., 2019b). This dataset provides a large collection of demonstrations from human players solving six different tasks in the sandbox game MineCraft. In addition to the human demonstrations the MineRL dataset also provides an OpenAI-Gym wrapper for MineCraft. The dataset includes demonstrations for the following tasks:

- navigating to target location following a compass,
- collecting wood by chopping trees,
- obtaining an item by collecting resources and crafting, and
- free play "survival" where the player is free to choose his own goal.

The demonstrations include the video showing the players' view (without user interface), the players' inventory at every time step and the actions performed by the player. We focus on the third task of obtaining a target item, namely a diamond. This task is very challenging as it is necessary to obtain several different resources and tools and has been the focus of a challenge (Guss et al., 2019a) at NeurIPS'19. By the end of this challenge no entry was able to obtain the diamond.

We show that our method is well suited for solving the task of obtaining the diamond, which can be decomposed into sub-tasks by reward redistribution after aligning successful demonstrations.

### A.5.2 RELATED WORK AND STEPS TOWARDS A GENERAL AGENT

In the following, we review two approaches Skrynnik et al. (2019); Scheller et al. (2020) where more details are available and compare them with our approach.

Skrynnik et al. (2019) address the problem with a TD based hierarchical Deep $Q$-Network (DQN) and by utilizing the hierarchical structure of expert trajectories by extracting sequences of meta-actions and sub-goals. This approach allowed them to achieve the *1st* place in the official NeurIPS'19 MineRL challenge (Skrynnik et al., 2019). In terms of pre-processing, our approaches have in common that both rely on frame skipping and action space discretization. However, they reduce the action space to ten distinct joint environment actions (e.g. *move camera & attack*) and treat inventory actions separately by executing a sequence of semantic actions. We aim at taking a next step towards a more general agent by introducing an action space preserving the agent's full freedom of action in the environment (more details are provided below). This allows us to avoid the distinction between item (environment) and semantic (inventory) agents and to train identically structured agents in the same fashion regardless of facing a mining, crafting, placing or smelting sub-task. Skrynnik et al. (2019) extract a sub-task chain by separately examining each expert trajectory and by considering the time of appearance of items in the inventory in chronological order. For agent training their approach follows a heuristic where they distinguish between collecting the item *log* and all remaining items. The *log*-agent is trained by starting with the *TreeChop* expert trajectories and then gradually injecting trajectories collected from interactions with the environment into the DQN's replay buffer. For the remaining items they rely on the expert data of *ObtainIronPickaxeDense* and imitation learning. Given our proposed sequence alignment and reward redistribution methodology we are able to avoid this shift in training paradigm and to leverage all available training data (*ObtainDiamond*, *ObtainIronPickaxe* and *TreeChop*) at the same time. In short, we collect all expert trajectories in one pool, perform sequence alignment yielding a common diamond consensus along with the corresponding reward redistribution and the respective sub-task sequences. Given this restructuring of the problem into local sub-problems with redistributed reward all sub-task agents are then trained in the same fashion (e.g. imitation learning followed by RL-based fine-tuning). Reward redistribution guarantees that the optimal policies are preserved (Arjona-Medina et al., 2019).

Scheller et al. (2020) achieved the *3rd* place on the official leader board following a different line of research and addressed the problem with a single *end-to-end* off-policy IMPALA (Espeholt et al.,

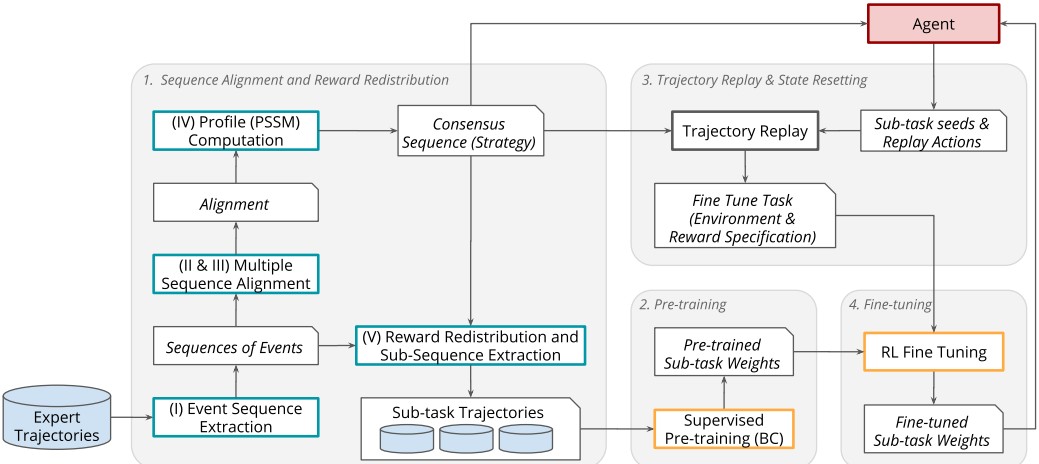

Figure A.9: Conceptual overview of our MineRL agent.

2018) actor-critic agent, again utilizing experience replay to incorporate the expert data (Scheller et al., 2020). To prevent catastrophic forgetting of the behavior for later, less frequent sub-tasks they introduce value clipping and apply CLEAR (Rolnick et al., 2019) to both, policy and value networks. Treating the entire problem as a whole is already the main distinguishable feature compared to our method. To deal with long trajectories they rely on a trainable special form of frame skipping where the agent also has to predict how many frames to skip in each situation. This helps to reduce the effective length (step count) of the respective expert trajectories. In contrast to the approach of (Scheller et al., 2020) we rely on a constant frame skip irrespective of the states and actions we are facing. Finally, there are also several common features including:

1. a supervised BC pre-training stage prior to RL fine-tuning,
2. separate networks for policy and value function,
3. independent action heads on top of a sub-sequence LSTM,
4. presenting the inventory state in a certain form to the agent and
5. applying a value-function-burn-in prior to RL fine-tuning.

### A.5.3 IMPLEMENTATION OF OUR ALGORITHM FOR MINECRAFT

The architecture of the training pipeline incorporates three learning stages:

- sequence alignment and reward redistribution
- learning from demonstrations via behavioral cloning (pre-training) and
- model fine-tuning with reinforcement learning.

Figure A.9 shows a conceptual overview of all components.

**Sequence alignment and reward redistribution.** First, we extract the sequence of states from human demonstrations, transform images into feature vectors using a standard pre-trained network and transform them into a sequence of consecutive state deltas (concatenating image feature vectors and inventory states). We cluster the resulting state deltas and remove clusters with a large number of members and merged smaller clusters. This results in 19 events and we map the demonstrations to sequences of events. These events correspond to inventory changes. For each human demonstration we get a sequence of events which we map to letters from the amino acid code, resulting in a sequence of letters. In Fig. A.10 we show all events with their assigned letter encoding that we defined for the Minecraft environment.

We then calculate a scoring matrix according to step (II) in Sec. 3 in the main document. Then, we perform multiple sequence alignment to align sequences of events of the top $N$ demonstrations, where shorter demonstrations are ranked higher. This results in a sequence of common events which

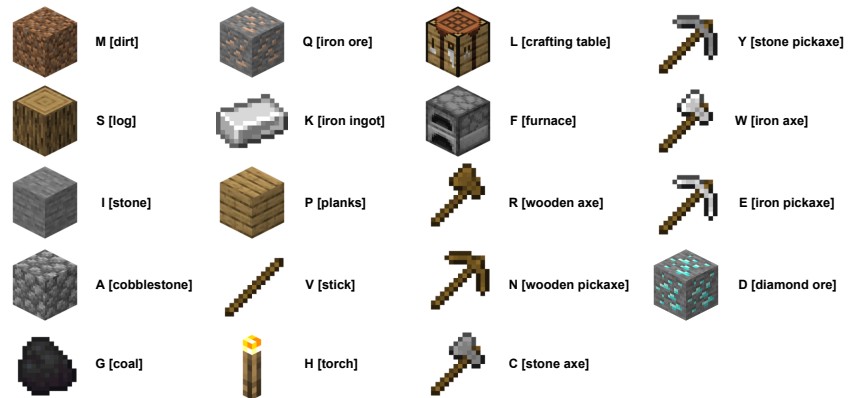

Figure A.10: Mapping of clusters to letters.

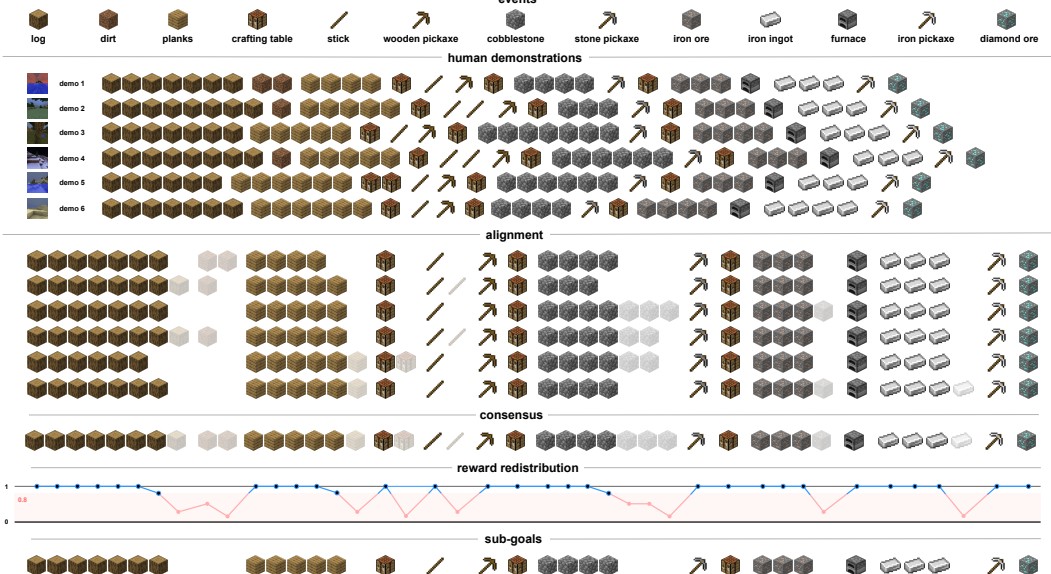

Figure A.11: Example of alignment and reward redistribution for demonstrations of `ObtainDiamond`. Thresholding the redistributed reward identifies sub-goals.

we denote as the consensus. In order to redistribute the reward, we use the PSSM model and assign the respective reward. Reward redistribution allows the sub-goal definition i.e. positions where the reward redistribution is larger than a threshold or positions where the reward redistribution has a certain value. In our implementation sub-goals are obtained by applying a threshold to the reward redistribution. The main agent is initialized by executing sub-agents according to the alignment. Figure A.11 shows how sub-goals are identified using reward redistribution.

**Learning from demonstrations via behavioral cloning.** We extract demonstrations for each individual sub-task in the form of sub-sequences taken from all demonstrations. For each sub-task we train an individual sub-agent via behavioral cloning.

**Model fine-tuning with reinforcement learning.** We fine-tune the agent in the environment using PPO (Schulman et al., 2018). During fine-tuning with PPO, an agent receives reward if it manages to reach its sub-goal.

To evaluate the performance of an agent for its current sub-goal, we average the return over multiple roll-outs. This gives us a good estimate of the success rate and if trained models have improved during fine tuning or not. In Fig. 5, we plot the overall success rate of all models evaluated sequentially from start to end.

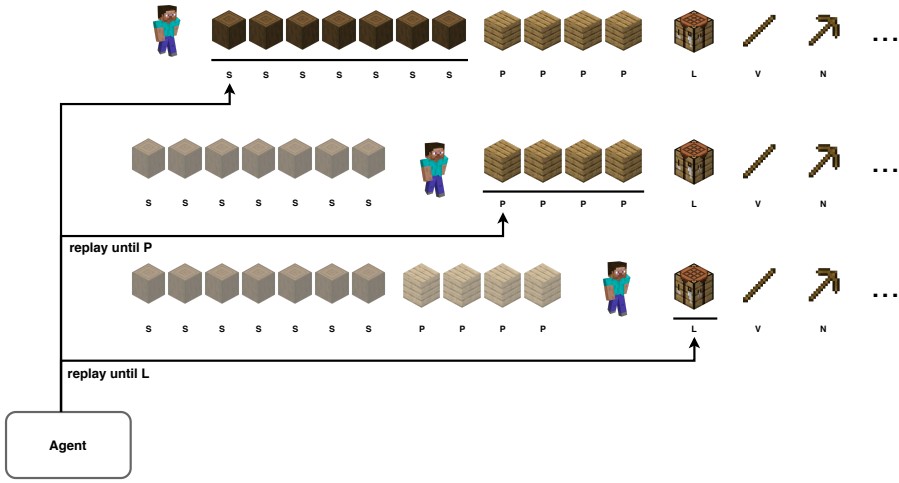

Figure A.12: Trajectory replay given by an exemplary consensus. The agent can execute training or evaluation processes of various sub-tasks by randomly sampling and replaying previously recorded trajectories on environment reset. Each letter defines a task. L (log), P (planks), V (stick), L (crafting table) and N (wooden pickaxe).

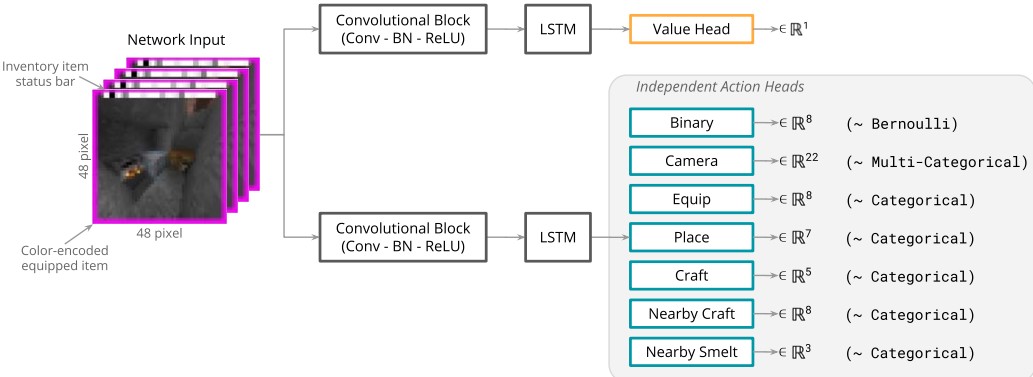

Figure A.13: Conceptual architecture of our MineRL policy and value networks.

In order to shorten the training time of our agent, we use trajectory replay and state resetting, similar to the idea proposed in (Ecoffet et al., 2019), allowing us to train sub-task agents in parallel. This is not necessary for the behavioral cloning stage, since here we can independently train agents according to the extracted sub-goals. However, fine-tuning a sub-task agent with reinforcement learning requires agents for all previous sub-tasks. To fine-tune agents for all sub-tasks, we record successful experiences (states, actions, rewards) for earlier goals and use them to reset the environment where a subsequent agent can start its training. In Fig. A.12, we illustrate a trajectory replay given by an exemplary consensus.

### A.5.4 POLICY AND VALUE NETWORK ARCHITECTURE

Figure A.13 shows a conceptual overview of the policy and value networks used in our MineRL experiments. The networks are structured as two separate convolutional encoders with an LSTM layer before the respective output layer, without sharing any model parameters.

The input to the model is the sequence of the 32 most recent frames, which are pre-processed in the following way: first, we add the currently equipped item as a color-coded border around each RGB frame. Next, the frames are augmented with an inventory status bar representing all 18 available inventory items (each inventory item is drawn as an item-square consisting of $3 \times 3$ pixels to the frame). Depending on the item count the respective square is drawn with a linearly interpolated

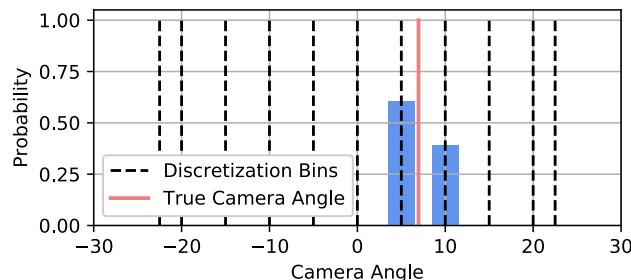

Figure A.14: Discretization and interpolation of camera angles.

gray-scale ranging from white (no item at all) to black (item count $> 95$). The count of 95 is the 75-quantile of the total amount of collected cobblestones and dirt derived from the inventory of all expert trajectories. Intuitively, this count should be related to the current depth (level) where an agent currently is or at least has been throughout the episode. In the last pre-processing step the frames are resized from $64 \times 64$ to $48 \times 48$ pixels and divided by 255 resulting in an input value range between zero and one.

The first network stage consists of four batch-normalized convolution layers with ReLU activation functions. The layers are structured as follows: Conv-Layer-1 (16 feature maps, kernel size 4, stride 2, zero padding 1), Conv-Layer-2 (32 feature maps, kernel size 4, stride 2, zero padding 1), Conv-Layer-3 (64 feature maps, kernel size 3, stride 2), and Conv-Layer-4 (32 feature maps, kernel size 3, stride 2). The flattened latent representation ($\in \mathbb{R}^{32 \times 288}$) of the convolution stage is further processed with an LSTM layer with 256 units. Given this recurrent representation we only keep the last time step (e.g. the prediction for the most recent frame).

The value head is a single linear layer without non-linearity predicting the state-value for the most recent state. For action prediction, two types of output heads are used depending on the underlying action distribution. The binary action head represents the actions *attack*, *back*, *forward*, *jump*, *left*, *right*, *sneak* and *sprint* which can be executed concurrently and are therefore modeled based on a *Bernoulli* distribution. Since only one item can be equipped, placed, or crafted at a time these actions are modeled with a *categorical* distribution. The equip head selects from *none*, *air*, *wooden-axe*, *wooden-pickaxe*, *stone-axe*, *stone-pickaxe*, *iron-axe* and *iron-pickaxe*. The place head selects from *none*, *dirt*, *stone*, *cobblestone*, *crafting-table*, *furnace* and *torch*. The craft head selects from *none*, *torch*, *stick*, *planks* and *crafting-table*. Items which have to be crafted nearby are *none*, *wooden-axe*, *wooden-pickaxe*, *stone-axe*, *stone-pickaxe*, *iron-axe*, *iron-pickaxe* and *furnace*. Finally, items which are smelted nearby are *none*, *iron-ingot* and *coal*. For predicting the camera angles (*up/down* as well as *left/right*) we introduce a custom action space outlined in Figure A.14. This space discretizes the possible camera angles into 11 distinct bins for both orientations leading to the 22 output neurons of the camera action head. Each bin holds the probability for sampling the corresponding angle as a camera action, since in most of the cases the true camera angle lies in between two such bins. We share the bin selection probability by linear interpolation with respect to the distance of the neighboring bin centers to the true camera angle. This way we are able to train the model with standard categorical cross-entropy during behavioral cloning and sample actions from this categorical distribution during exploration and agent deployment.

### A.5.5 IMITATION LEARNING OF SUB-TASK AGENTS

Given the sub-sequences of expert data separated by task and the network architectures described above we perform imitation learning via behavioral cloning (BC) on the expert demonstrations. All sub-task policy networks are trained with a cross-entropy loss on the respective action distributions using stochastic gradient decent with a learning rate of 0.01 and a momentum of 0.9. Mini-batches of size 256 are sampled uniformly from the set of sub-task sequences. As we have the MineRL simulator available during training we are able to include all sub-sequences in the training set and evaluate the performance of the model by deploying it in the environment every 10 training epochs. Once training over 300 epochs is complete we select the model checkpoint based on the total count

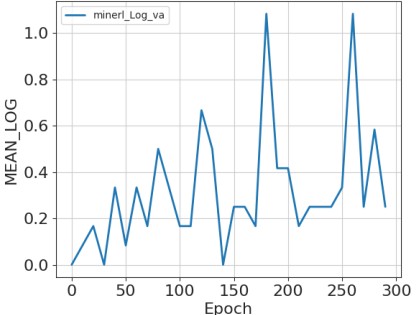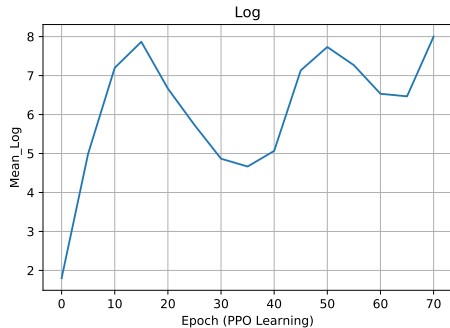

Figure A.15: Average number of logs collected during training: left: Behavioral Cloning, Right: PPO Training

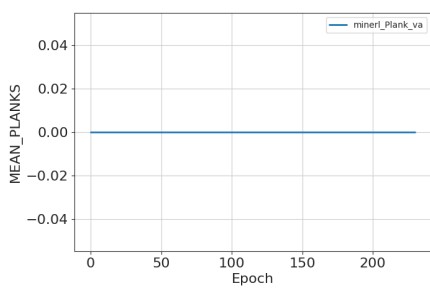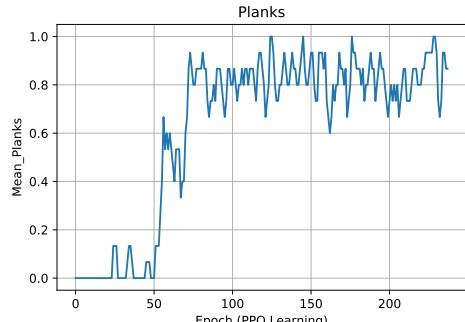

Figure A.16: Average number of planks crafted during training: left: Behavioral Cloning, Right: PPO Training

of collected target items over 12 evaluation trials per checkpoint. Due to presence of only successful sequences, the separate value network is not pre-trained with BC.

### A.5.6 REINFORCEMENT LEARNING ON SUB-TASK AGENTS

After the pretraining of the Sub-Task agents, we further fine tune the agents using PPO in the MineRL environment. The reward is the redistributed reward given by Align-RUDDER. The value function is initialized in a burn-in stage prior to policy improvement where the agent interacts with the environment for 50k timesteps and only updates the value function. Finally, both policy and the value function are trained jointly for all sub-tasks. All agents are trained between 50k timesteps and 500k timesteps. We evaluate each agent periodically during training and in the end select the best performing agent per sub-task. A.15 - A.19 present evaluation curves of some sub-task agents during learning from demonstrations using behavioral cloning and learning online using PPO.

### A.6 REPRODUCING THE ARTIFICIAL TASK RESULTS

The code to reproduce the results and figures of both artificial tasks is provided as supplementary material. The README contains step-by-step instructions to set up an environment and run the experiments. By default, instead of using 100 seeds per experiment only 10 are used in the demonstration code.

Finally, a video showcasing the MineCraft agent is also provided as supplementary material.

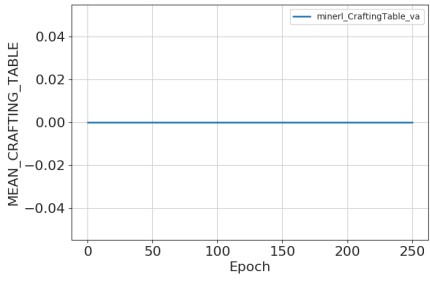 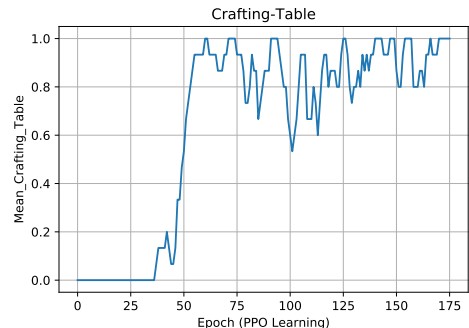

Figure A.17: Average number of table crafted during training: left: Behavioral Cloning, Right: PPO Training

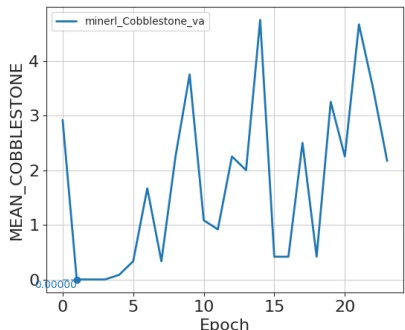 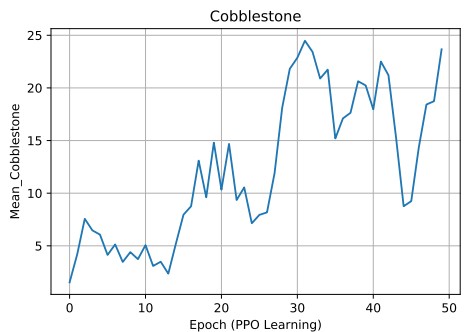

Figure A.18: Average number of stone collected during training: left: Behavioral Cloning, Right: PPO Training

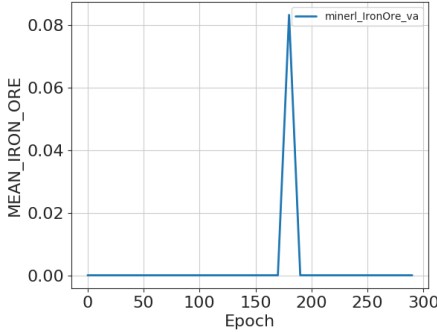 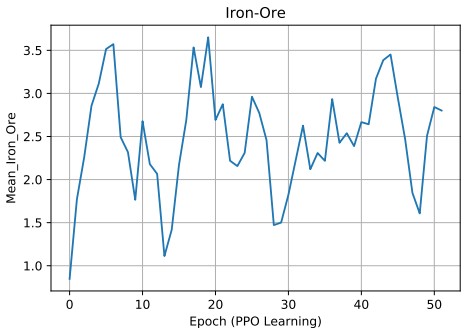

Figure A.19: Average number of iron-ore collected during training: left: Behavioral Cloning, Right: PPO Training

