# OpenReview forum: "Align-RUDDER: Learning From Few Demonstrations by Reward Redistribution"
_ICLR.cc/2021/Conference — Reject_

### Official Review · AnonReviewer4 · 2020-10-27
**Review for Align-Rudder**

**Rating:** 5
**Confidence:** 4

**Review:**

Summary:
The paper presents a method that learns from few demonstrations by computing a reward function coming from an alignement score between agent trajectories and expert demonstrations. The alignement score is inspired by methods used in bioinformatics called profile models.

Comments:
I think the paper could be better organise and not focus so much on the RUDDER framework. You could simply look at this problem from the angle of training an RL agent to produce aligned trajectories with the trajectories given by the expert where the local reward at time t is given by the profile models: g((s,a)_{0:t}) - g((s,a)_{0:t-1}).  This reward simply represents how much I deviated or not from the expert trajectories between time t and time t-1. Therefore presenting the method as a trajectory-matching imitation learning method would be way simpler for the reader. Too much emphasis is given to RUDDER and the message of the paper is somehow diluted whereas the algorithm could be simply presented. Indeed almost one page is dedicated to present results on reward distribution whereas I would like, for instance, to have more information on how the events are computed and not be referred to a paper. Indeed understanding how events are computed is crucial because they define the intrinsic metric used to compute distances between trajectories. This has been for a long time the main problem of trajectory-matching methods compared to distribution matching method. Therefore it seems crucial to develop this point.
It seems also that the number of events should not be too large for the alignement method to work. It would be nice to have more information on that specific problem. In addition such alignement methods may have problems in stochastic environments unless the events are really well defined. Could the authors expand on this (deterministic vs stochastic environments)?
Finally I find that the experimental section could be more exhaustive qualitatively and quantitatively. More environments (more or less stochastic) are needed, showing an experiment that pushes the algorithm in terms of number of events to see how it behaves, having an experiment showing different type of events and how it impact the performance. In general, as a practitioner, I would like to have a better understanding of the method  and in particular its robustness that is why I am asking for those experiments.

Rating: I think the paper could be simplify by focusing on solving the problem of learning from demonstrations by minimising a distance between expert and agent trajectories with an alignment score. This can be framed as trajectory matching imitation learning. In addition, a more extended experimental section is needed to show the limits of the method. At the moment, it is not ready for publication.

---

> ### Author Response · Authors · 2020-11-18
> **Response to Reviewer 4**
>
> Thank you for your review and the suggestions on how to present our method better, which allowed us to improve the paper a lot.
>
> * “not focus so much on the RUDDER framework ” & “Too much emphasis is given to RUDDER”: We would like to keep the theoretical guarantees, which are based on the RUDDER framework.
>
> * “presenting the method as a trajectory-matching imitation learning”: The reviewer is right, however we would like to cast our approach as “strategy following”, to avoid confusion with the original state-action trajectories.  First, we have a trajectory of events since state-actions are clustered to obtain events.  Secondly, the profile only focuses on events that agree to each other in the demonstrations while events that are not matched are not followed. We include the reviewer’s suggestion in the new version and write “Align-RUDDER can be cast as following the underlying strategy, which has been extracted by the profile model, therefore is a strategy-matching imitation learning method”.
>
> * “more information on how the events are computed”: Sorry for this shortcoming. We have rewritten the paragraph “Defining events” which is dedicated to this question. In general, events can be the original state-action pairs, clusters thereof, or other representations of state-action pairs, e.g. indicating changes of inventory, health, energy, skills etc. We define events as a cluster of states or state-actions. A sequence of events is obtained from a state-action sequence by substituting states or state-actions by their cluster identifier. We write “... For a given policy that provides state-action sequences, a successor representation supplies a similarity matrix that indicates the connectivity of two states. As given policy, we use the demonstrations combined with state-action sequences generated by a random policy. Also “successor features” (Barreto et al., 2017) may be used. Subsequently, we use similarity-based clustering like affinity propagation (AP) (Frey & Dueck, 2007). For affinity propagation the similarity matrix does not have to be symmetric and the number of clusters not to be known. Each cluster corresponds to an event.“
> For more clarity, we have also added more exemplary cluster plots for the toy examples in the appendix to better visualize the clusters.
>
> * “It seems also that the number of events should not be too large”: We write now: “If there are too many events, then for two demonstrations only few events can be matched, therefore aligned demonstrations cannot be distinguished from random alignments. This effect is known in bioinformatics as ‘Inconsistency of Maximum Parsimony’ (Felsenstein, 1978).”
> “Inconsistency of Maximum Parsimony (Felsenstein, 1978)”: Assume evolutionary close related sequences A and B, which have few amino acids in common, because B has a high mutation rate. Then it is possible that A has more amino acids in common with C, even if C is more remotely related to A than B. The effect appears because related sequences have only few matching amino acids. Even unrelated sequences might have more matching amino acids than the evolutionary related sequences A and B.
>
> * “methods may have problems in stochastic environments” & “experimental section could be more exhaustive qualitatively and quantitatively”: The reviewer is right. The more stochastic the environment is, the harder is the clustering of events and the harder is the alignment. Therefore, we extended the toy experiments by varying the randomness. We added Imitation Learning via Regularized Behavioral Cloning (SQIL) as an additional baseline to the paper which was already mentioned to be suited to learn from few demonstrations. We also  started experiments with Policy Optimization from Demonstrations (POfD, Kang et al. , 2018). Preliminary experiments for POfD did not converge even for 100 demonstrations but the experiments are still running. More results on stochastic environments are presented now in the appendix.
>
> * “experimental section could be more exhaustive qualitatively and quantitatively”:  We also included baselines for the minecraft experiment by comparing to methods from the NeurIPS challenge. We wrote: “Align-RUDDER was not evaluated during the challenge, therefore may have advantages. Though it did not receive the intermediate rewards provided by the challenge that hint at sub-tasks.”

---

### Official Review · AnonReviewer1 · 2020-10-28
**Review for paper "Align-RUDDER: Learning From Few Demonstrations by Reward Redistribution"**

**Rating:** 6
**Confidence:** 3

**Review:**

[Summary]

Paper proposes to attack the challenging problem of RL with sparse feedback by leveraging a few demonstrations and learnable reward redistribution. The redistributed reward is computed by aligning the key events (a set of clustered symbols) to the demonstrations via PSSM-based seq matching. Experiments on two artificial tasks and a Minecraft task demonstrate that the presented method performs advantageously than two baselines (DQfD and BC+Q learning).

[Strength]

+) The authors motivate their central research problem well with a rich background ranging from the canonical RL context of credit assignment to the biological seq alignment, which helps enhances the rationale of the proposed method.

+) The paper is overall clear and well-written. The proposed approach is technically sound and I cannot find any issue.

+) Reused from the previous RUDDER paper though, the author provides a comprehensive theoretical analysis on how the proposed reward redistribution could not alter the optimality of the original problem, and how can these results adapt to the new configuration with demonstrations.

[Weakness]

Having said those above, indeed I feel this submission could be revised from several angles. Some of them are minor while others could be crucial to the acceptance in this round.

-) (major) The main claim of this paper is a "reward redistribution" strategy that utilizes expert demonstrations. Compared to RUDDER, the redistributed reward is computed by weighting the return with the differences between the trajectory similarities of two consecutive states. However, after reading the paper, I'm still not confident enough about the rationale of this practice. An even more critical point is that I really doubt the validity of redistributing the return by merely comparing it with the demonstration. Is it still reasonable to judge whether a specific time step is more important than the others when the trajectory of the learner may include events that do not happen in the demonstrations? Almost all the existing work is to use demonstrations for shaping the training either directly (as shaped rewards) or indirectly (as constraints) since the demonstrations may not cover the same state visitation of the learner's rollouts, and therefore, could be problematic to explain away how should the return be distributed. The authors are expected to clarify why their redistribution strategy is reasonable given the possible event set misalignment between expert and learner, and also why choosing the difference between similarity as the key metric of redist.

-) (minor) The authors provide a rich literature review of many seminal research works of combing imitation learning and reinforcement learning. However, only very few of them are compared in the experiments. Also, only two artificial tasks are evaluated for comparison. The authors are encouraged to add more baselines to their method list (e.g. POfD, THOR, etc) and scale up to standard benchmark (at least on the Minecraft domain).


[Suggestions&Questions]

(1) Clarify the rationale of redistributing the reward with the differences of similarity of two consecutive states.

(2) Is it still reasonable to redistribute the reward merely based on the demonstrations when novel states appear in the learner's trajectories?

(3) Add more baselines (at least POfD or its variants) and compared them on standard benchmarks, e.g. the Minecraft domain.

[Post-rebuttal]

I have read through all the other reviews and the rebuttal. Would like to thank the authors for their efforts in improving this submission. I do believe most of my concerns have been addressed. However, the concern on some possibly confusing technical details remains. The authors are expected to further revise their paper to make it more self-explained.

---

> ### Author Response · Authors · 2020-11-18
> **Response to Reviewer 1**
>
> Thank you for your helpful review and the suggestions & questions which were very useful to improve the paper.
>
> * Question (2) about novel states: As described in the paragraph “Learning methods according to (Arjona-Medina et al. 2019)”, the reward redistribution is the input to a subsequent learning algorithm. These methods further redistribute the reward to states / events that are encountered the first time during learning. These methods would also work, if reward is only given at sequence end. However, reward redistribution gives the reward earlier. To make this clearer, we added: “The redistributed reward serves as reward for a subsequent learning method, which can be Type A, B, and C as described in (Arjona-Medina et al. 2019).” We now more explicitly mention the subsequent learning methods Q-learning and Proximal Policy Optimization (PPO).
> We can already redistribute reward without going to the sequence end and knowing the final return. At the sequence end, we give a correction reward which corrects for wrong redistributions. Misalignments would not allow to redistribute the reward (no agreement of events) and the reward stays at sequence end, where the redistribution is corrected.
>
> * Question (1) concerning differences of similarity of two consecutive states: The alignment score is return-decomposition function g. Like with LSTM networks we just compute R_{t+1}=g((s,a)_{0:t}) - g((s,a)_{0:t-1}) for the redistributed reward. The sum is a telescope sum: \sum_{t=0}^T  R_{t+1} = g((s,a)_{0:T}) - g((s,a)_{0:-1}) = g((s,a)_{0:T}). To justify why the redistribution strategy is reasonable, we now write first: “where g is the return decomposition function, which is an LSTM model that predicts the return of the episode.” Then we write later: “The redistributed reward of a new sequence is the difference of the scores of consecutive sub-sequences when aligned to the profile model, where the alignment score is the return-decomposition function g. This difference indicates how much of the return is gained or lost by a sequence element, which is the main idea of return decomposition: g((s,a)_{0:t}) - g((s,a)_{0:t-1}).”
>
> * Disclaimer: Reward redistribution keeps the optimal policies. The reward redistribution by differences between the trajectory similarities is causal, that means, the reward depends only on the past. Therefore, a reward redistribution might misguide the learning process but the optimal policies are still the same.
>
> * Question (3) concerning more baselines and experiments: We extended the toy experiments by varying the randomness. We added Imitation Learning via Regularized Behavioral Cloning (SQIL) as an additional baseline to the paper which was already mentioned to be suited to learn from few demonstrations. We also  started experiments with Policy Optimization from Demonstrations (POfD, Kang et al. , 2018). Preliminary experiments for POfD did not converge even for 100 demonstrations but the experiments are still running. We also included baselines for the minecraft experiment by comparing to methods from the NeurIPS challenge. We wrote: “Align-RUDDER was not evaluated during the challenge, therefore may have advantages. Though it did not receive the intermediate rewards provided by the challenge that hint at sub-tasks.” Some further results are now presented in the appendix.

---

### Official Review · AnonReviewer3 · 2020-10-28
**RUDDER is an algorithm published in 2019 that tries to address challenges posed by delayed rewards in reinforcement learning. This paper extends RUDDER in several ways. It applies RUDDER to learning from a small number of demonstrations. It has to replace LSTM with profile models used in sequence matching in biology because the number of human demonstrations available for learning is small. The results on several domains are encouraging.**

**Rating:** 7
**Confidence:** 4

**Review:**

This paper presents strong research and is very well written. The authors managed to put many technical details in a very limited space. The quality of writing is far above the average. I am very positive about this paper, but I also have a few concerns that don't allow me to give a very high score in my initial review.

Sub-tasks are an important element of the approach, but the authors did not even cite the most classical papers on hierarchical reinforcement learning. The two standard and well-known approaches are:

Dietterich, T.G., 2000. Hierarchical reinforcement learning with the MAXQ value function decomposition. Journal of artificial intelligence research, 13, pp.227-303.

Sutton, R.S., Precup, D. and Singh, S., 1999. Between MDPs and semi-MDPs: A framework for temporal abstraction in reinforcement learning. Artificial intelligence, 112(1-2), pp.181-211.

These papers are highly cited, and they were extended in various ways, and many researchers tried to learn the sub-tasks automatically - among other things. I have to say that I am quite shocked that the authors did not cite those two very classical references (or at least one of them). This paper should clearly state how the ideas presented here relate to the MaxQ algorithm, and the temporally extended actions (options) proposed by Sutton et al. The sub-tasks presented here clearly resemble Sutton's options. This relationship should be discussed. In particular, learning Sutton's options was not a trivial task in the past. The authors should explain why sub-task learning seems to be trivial in this paper. What is the main reason or the main assumption that allows for that? What are the main simplifying assumptions made in this paper that allow for easy sub-task learning?

Figure 1 is very informative. I would ask however what would happen if the agent lost the key? The blue line should go down I assume. It would be good to see what the RUDDER algorithm would do when negative things (like lost keys) happen.

Section 3 assumes that demonstrations can be aligned. This means that if the goal can be achieved using two alternative paths of similar quality, the methods proposed here cannot be applied. The authors should say a bit more about this limitation. Could it be mitigated? If not so, this limitation mustn't be hidden.

A related question: the environment can be highly stochastic. Assume that an agent takes action a, and ends up in one of the states s1, s1, or s2 with the same probability. Also, assume that the goal state can be reached from each of these 3 states, but one has to follow a different path from each of these states. That is, all the paths that go from s1, s2, and s3 to the goal state are disjoint. Would this algorithm cope with such a situation?

The following statement is unclear to me: "In our setting the states do not have to be time-aware for
ensuring an MDP but the unobserved used-up time introduces a random effect."

I am not certain about correctness of this "Demonstration sub-sequences between sub-goals are considered as demonstrations for the sub-tasks." The agent may pick the key up, but it can drop it a few time steps later, and then pick it up again. This would not be a useful demonstration for the sub-task I think. How does the algorithm deal with the goals that be done and undone many times?

There is potentially significant novelty in this well-written paper. I would prefer to give a stronger recommendation to accept it, but I am disappointed that references to the key literature on hierarchical reinforcement learning are missing.

---

> ### Author Response · Authors · 2020-11-18
> **Response to Reviewer 3 (Part 1)**
>
> Thank you for a very insightful review that helped us to improve our paper, in particular concerning references to hierarchical reinforcement learning literature, which were missing.
>
> * “cite the most classical papers on hierarchical reinforcement learning”: The reviewer is right - sorry for this shortcoming. We now cite the mentioned papers and several others on hierarchical reinforcement learning, therefore we extended the related work part by a paragraph on hierarchical reinforcement learning.
> We also now state clearly that Align-RUDDER is a method to identify options, sub-goals and sub-task and automatically assigns rewards to them, therefore is a method for hierarchical reinforcement learning.
> The reason for our shortcoming was that reward redistribution was originally not designed for hierarchical RL. Hierarchical RL appeared automatically during our work.
>
> * “what would happen if the agent lost the key?”: The reviewer is absolutely right. The blue line will go down. Consequently, negative reward is redistributed: getting the key relates to a positive reward while losing it relates to a negative reward of the same amount. This effect was described and observed in the RUDDER paper, e.g. in the videos that have been distributed with the RUDDER paper.
>
> * “two alternative paths of similar quality”: We addressed this issue by writing: “Then, a guiding tree is produced via hierarchical clustering sequences, according to their pairwise alignment scores. Demonstrations which follow the same strategy appear in the same cluster in the guiding tree. Each cluster is aligned separately via MSA to address different strategies.” If there are two alternative paths then this would be reflected in the guiding tree and the two paths would be aligned separately.
> To improve our explanation, we add in the new version at the beginning of Section 3: “However, if there is not an underlying strategy, then the alignment will fail. Events will not receive a high redistributed reward and the reward is given at sequence end, when the redistributed reward is corrected. Then our method does not give an advantage, since the assumption of an underlying strategy is violated.”
> Later we write in “(III) Multiple sequence alignment (MSA)”: “However, if there is not a cluster of demonstrations, then the alignment will fail.”
>
> * “states s1, s1, or s2 ”: A good point. Align-RUDDER has two solutions to this problem. Solution I: By defining events. The states s1, s2 and s3 may be clustered into the same event, since the states that follow these states are similar to each other. Then the three states are the same event. Solution II: By the alignment. The states are clustered in different events, therefore they might not be aligned. However, the alignment can still work, since other events can be aligned.  In the worst case there is no solution: the alignment fails, no high reward is redistributed and the reward is given at sequence end, when the redistributed reward is corrected. Then Align-RUDDER does not give an advantage.
>
> * “the environment can be highly stochastic”: We conducted additional experiments with different levels of stochasticity in order to evaluate the robustness of Align-RUDDER to stochastic environments. Some further results on stochastic environments are now presented in the appendix.
>
> * More experiments: We extended the toy experiments by using environments with different stochasticity. We added Imitation Learning via Regularized Behavioral Cloning (SQIL) as an additional baseline to the paper which was already mentioned to be suited to learn from few demonstrations. We also started experiments with Policy Optimization from Demonstrations (POfD, Kang et al. , 2018). Preliminary experiments for POfD did not converge even for 100 demonstrations but the experiments are still running. We also included baselines for the minecraft experiment by comparing to methods from the NeurIPS challenge. We wrote: “Align-RUDDER was not evaluated during the challenge, therefore may have advantages. Though it did not receive the intermediate rewards provided by the challenge that hint at sub-tasks.”
> Some further results are now presented in the appendix.
>
> * “do not have to be time-aware for ensuring an MDP”: Thanks for pointing that out: it is an error. We corrected it to: "In our setting the states do not have to be time-aware for ensuring stationary optimal policies but the unobserved used-up time introduces a random effect.". We assumed that the states are time-aware to assure stationary optimal policies for finite time horizon MDPs.

---

> > ### Comment · AnonReviewer3 · 2020-11-23
> > **After reading authors' response**
> >
> > Thank you for addressing our questions. I am satisfied with your answers, and I still think that this is a very good piece of work. I don't have any other questions to the authors, and I am admiring them for the amount of work that they did in a very short time to improve the paper and to write their response.

---

> ### Author Response · Authors · 2020-11-18
> **Response to Reviewer 3 (Part 2)**
>
> * "Demonstration sub-sequences between sub-goals are considered as demonstrations for the sub-tasks.": An agent picks up a key, then drops it, and picks it up again. We consider picking up the key the second time as a different sub-task, since it is at a different position in the alignment. We must stay agnostic, since we cannot know whether picking the key a second time is the same as picking it up the first time, e.g. the second time the key is nearby.  We observed this difference in the minecraft experiment: mining the first plank requires walking towards the wood while the second plank can be mined directly.

---

### Official Review · AnonReviewer2 · 2020-10-28
**Relevant interdisciplinary extension of prior work, good performance, slightly limited evaluation**

**Rating:** 7
**Confidence:** 4

**Review:**

The submission proposes to extend reward redistribution methods from RL (RUDDER) to learning from demonstration. The principal contribution of the submission is a method for using low numbers of demonstrations.
Essentially, the redistribution mechanism is adapted to switch from training neural networks to clustering and sequence alignment methods. The approach is shown to work well in toy tasks in multi-room navigation scenarios and a final sparse reward task from the MineRL competition.

The paper is clearly written and the introduction, reasoning behind the novel method are succinct and prior work and new contributions are clearly separated. The proposed method performs very well and the use of sequence alignment, scoring and clustering tools with prior background in bioinformatics is a good example for interdisciplinary work.

The experimental evaluation is limited with 2 toy tasks with low-dimensional observations (with relevant baselines) and a more complex high-dimensional task evaluating a strongly handcrafted adaptation of the proposed method (without strong baselines). Solving the sparse MineRL task is still impressive but it remains partially ambiguous which aspects of the now adapted algorithm and crafted event space contribute most.

Regarding related work, there exists a considerable amount of work addressing one or few shot learning from demonstration. All with their own requirements which will partially render them inapplicable to the tasks but they could nonetheless be linked under related work for a more complete picture.

The main limitation of the paper is the experimental evaluation for previously mentioned reasons. There are a couple of ways to strengthen the part:
One could add stronger baselines for the MineRL task. There is a lot of introduced structure: small set of events, independent training for events and finally use of the consensus strategy for switching between individual policies. A possible baseline would be to use the manually chosen set of events, train policies for all possible events independently and train high-level controller to switch between these policies and solve the task. This baseline would use the same information about the event space and independent training, but would need to train more agents and also the high-level controller. In the current text, this is hinted at but not actually executed.
One could also add additional ablations on the performance in the MineRL task. Evaluation of the policies without PPO fine-tuning or performance during training would be helpful. Similarly, if accepted the additional space could be used to show success in the final task during training. An option would be to port figure A.7 to the main paper.


(Disclaimer: I have reviewed a previous submission of this work. My previous review was positive and had only few requests - most of which are addressed.)

---

> ### Author Response · Authors · 2020-11-18
> **Response to Reviewer 2**
>
> Thank you for a very elaborate and profound review and suggestions that helped us to improve our paper.
>
> * “The experimental evaluation is limited”: We extended the toy experiments by varying the randomness. We added Imitation Learning via Regularized Behavioral Cloning (SQIL) as an additional baseline to the paper which was already mentioned to be suited to learn from few demonstrations. We also  started experiments with Policy Optimization from Demonstrations (POfD, Kang et al., 2018). Preliminary experiments for POfD did not converge even for 100 demonstrations but the experiments are still running.
> We also included baselines for the minecraft experiment by comparing to methods from the NeurIPS challenge. We wrote: “Align-RUDDER was not evaluated during the challenge, therefore may have advantages. Though it did not receive the intermediate rewards provided by the challenge that hint at sub-tasks.”
>
> * Concerning “strongly handcrafted”: We tried to avoid handcrafting for minecraft as much as possible. (i) We have chosen a particular representation for the minecraft states comprising the visual input and the inventory. (ii) We have chosen architectures for the neural networks like the convolutional neural network for the visual input. (iii) We have chosen the selection criteria for the demonstrations that were aligned: we selected the 10 shortest demonstrations that obtained the diamond. All other procedures were done automatically and are the same in the toy experiments and in minecraft: like clustering of the differences of states to define the events, removing rare events, and the alignment.
>
> * “Contribute most”: We are working on ablation studies but will not finish within the rebuttal time.
>
> * Related work “addressing one or few shot learning from demonstration”:  We included more references to few shot learning from demonstrations beyond the already cited methods Duan et al., 2017; Finn et al., 2017; Zhou et al., 2020.
>
> * Baseline via “manually chosen set of events”: The reviewer is right, this would be an interesting study but it would be computationally very demanding. We required months to train all the agents. It must be done iteratively since later agents are based on the success of previous agents. Currently this is infeasible with our compute infrastructure.
>
> * “performance during training would be helpful”: We produced plots for the performance during training in the appendix.
>
> * “port figure A.7 to the main paper”: Thanks for this hint. We will follow the suggestion of the review and move Figure A 7 to the main paper. In particular, it shows the performance for BC, which is without fine tuning via PPO.

---

> > ### Comment · AnonReviewer2 · 2020-11-24
> > **Feedback**
> >
> > Thank you for the quick feedback. This further supports my opinion of the paper and clarifies a couple of aspects. In particular, the additional baselines (and hopefully ablations for the final paper) are contributing to make a strong case for this work!

---

### Author Response · Authors · 2020-11-18
**Thank you for your feedback!**

We thank all reviewers for their time and for their constructive feedback. It helped us a lot to improve our paper. We hope to answer all questions and provide clarifications in individual responses to the respective reviewers. Further, we uploaded a rebuttal revision of our paper incorporating your sound suggestions. Concretely, (i) we included new baseline methods for the toy experiment, (ii) we included 7 additional experiments with different stochasticity values of the environment, and (iii) we included baselines for minecraft.

---

### Author Response · Authors · 2021-02-04
**Align-RUDDER is a variant of RUDDER and has its theoretical guarantees. Other variants are obtained by replacing the LSTM by GRUs, standard RNNs, ARMA (autoregressive models), or sequence analysis models like a profile model.**

We would like to take this opportunity to thank the reviewers again for their time and positive (7, 7, 6, 5) outlook towards our work. We addressed all the comments made by the reviewers during the rebuttal phase and the constructive feedback has helped improve our manuscript a lot.


Further, as authors, we disagree strongly with the notion that Align-RUDDER is not much related to RUDDER. Reward redistribution is the framework, the tool of reward redistribution (LSTM in RUDDER) is exchanged with sequence alignment in Align-RUDDER. To assign credit, RUDDER identifies steps in the Q-function and makes learning faster by reducing delay in reward. Similarly, Align-RUDDER identifies the largest steps in the Q-function via relevant events determined by the alignment profile model. The resulting reward redistribution is the same as RUDDER. As a result, Align-RUDDER fits into the RUDDER framework and has similar theoretical guarantees.


We further disagree that Align-RUDDER can be framed as imitation learning. Imitation learning aims at learning a behaviour close to the data generating policy by matching the trajectories of single demonstrations. In contrast, Align-RUDDER does not try to match single trajectories but tries to identify relevant events that are shared among successful demonstrations. Therefore, Align-RUDDER excels in complex problems like MineCraft with few demonstrations, where trajectory matching fails.

---

### Decision · Program_Chairs · 2021-01-07
**Final Decision**

**Decision:**

Reject

**Comment:**

The reviewers appreciated that the paper was clear and well written. They also appreciated that the paper has been largely improved during the discussions. The results seem to support the claim and the experiments on Minecraft are convincing.

Yet, the reviewers had some important concerns. First the focus on RUDDER seems too strong and the method doesn't seem to be that much related to RUDDER. Presenting the work as a trajectory matching method seems more appropriate. In addition, the authors support their choice of referring to RUDDER because it comes with theoretical guarantees. But RUDDER's guarantees come from the usage of a modified LSTM while Align-RUDDER doesn't use an LSTM.

The hierarchical approach was also questioned as the way to switch between different sub-policies is not very well explained in the paper. Baselines wrt to the switching method could not be provided. Similarly, the structure of the Minecraft task seems to be used heavily to define the hierarchy and meta-planning, so more baselines (with less structured tasks) were requested.

The method also suffers from scalability issues as the authors acknowledge that if the number of events grows, they would need to downsample the events so as to apply their method.